# Conversion of dietary inositol into propionate and acetate by commensal *Anaerostipes* associates with host health

Thi Phuong Nam Bui [1,2 ✉], Louise Mannerås-Holm[3], Robert Puschmann[4,5], Hao Wu[3,6], Antonio Dario Troise[7], Bart Nijsse[1], Sjef Boeren [8], Fredrik Bäckhed[3,9,10], Dorothea Fiedler [4,5] & Willem M. deVos [1,11 ✉]

We describe the anaerobic conversion of inositol stereoisomers to propionate and acetate by the abundant intestinal genus *Anaerostipes*. A inositol pathway was elucidated by nuclear magnetic resonance using [$^{13}$C]-inositols, mass spectrometry and proteogenomic analyses in *A. rhamnosivorans*, identifying 3-oxoacid CoA transferase as a key enzyme involved in both 3-oxopropionyl-CoA and propionate formation. This pathway also allowed conversion of phytate-derived inositol into propionate as shown with [$^{13}$C]-phytate in fecal samples amended with *A. rhamnosivorans*. Metabolic and (meta)genomic analyses explained the adaptation of *Anaerostipes* spp. to inositol-containing substrates and identified a propionate-production gene cluster to be inversely associated with metabolic biomarkers in (pre)diabetes cohorts. Co-administration of myo-inositol with live *A. rhamnosivorans* in western-diet fed mice reduced fasting-glucose levels comparing to heat-killed *A. rhamnosivorans* after 6-weeks treatment. Altogether, these data suggest a potential beneficial role for intestinal *Anaerostipes* spp. in promoting host health.

[1] Laboratory of Microbiology, Wageningen University, Wageningen, The Netherlands. [2] Caelus Pharmaceuticals, Zegveld, The Netherlands. [3] The Wallenberg Laboratory, Department of Molecular and Clinical Medicine, Institute of Medicine, Sahlgrenska Academy, University of Gothenburg, Gothenburg, Sweden. [4] Leibniz-Forschungsinstitut für Molekulare Pharmakologie, Berlin, Germany. [5] Institute of Chemistry, Humboldt-Universität zu Berlin, Berlin, Germany. [6] Human Phenome Institute, Fudan University, Shanghai, China. [7] Proteomics & Mass Spectrometry Laboratory, ISPAAM, National Research Council, Portici, NA, Italy. [8] Laboratory of Biochemistry, Wageningen University, Wageningen, The Netherlands. [9] Novo Nordisk Foundation Center for Basic Metabolic Research, Faculty of Health and Medical Sciences, University of Copenhagen, Copenhagen, Denmark. [10] Region Västra Götaland, Sahlgrenska University Hospital, Department of Clinical Physiology, Gothenburg, Sweden. [11] Human Microbiome Research Program, Faculty of Medicine, University of Helsinki, Helsinki, Finland. ✉email: nam.bui@wur.nl; willem.devos@wur.nl

Microbes inhabiting the human intestinal tract form a large and dynamic microbial community that infers important health benefits to the host[1]. One of the major functions of the intestinal microbes is the biosynthesis of absorbable metabolites conferring effects on intestinal epithelia and beyond[2]. Most studied are the short chain fatty acids (SCFAs) that are known to signal to various G-protein coupled receptors, have epigenetic effects and provide both metabolic and immune signalling[3,4]. Acetate is the most abundant faecal SCFA and has been suggested to affect body weight control, insulin sensitivity and control of appetite[5,6]. Propionate and butyrate are the main other SCFAs and not only signal to various receptors but also have epigenetic effects by inhibiting histone deacetylase activity[7]. Studies in human and model animals showed propionate to decrease liver lipogenesis, hepatic and serum cholesterol levels[8], to reduce satiety[9,10], and to induce apoptosis[11,12]. Three propionate synthesis pathways have been identified in human intestinal microbes, including the succinate, propanediol and acrylate pathways[13]. Among these, the succinate pathway appeared as the most abundant route and has been mainly observed to operate in Bacteroidetes, Negativicutes and Verrucomicrobia. The propanediol pathway was found to be widely spread within intestinal anaerobes, including some abundant bacteria such as *Ruminococcus obeum* (now *Blautia obeum*) and *Eubacterium hallii* (now known as *Anaerobutyricum* spp)[14,15]. The acrylate pathway was the least abundant with limited distribution in the gut community, including *Coprococcus catus* and *Megasphaera elsdenii*[13].

Butyrate is another important SCFA that is vital to maintain healthy intestinal barrier functions[16]. Increasing evidence reveals a strong inverse relation between butyrate-producing bacteria and potential pathogenesis[17–19]. Butyrate-producing bacteria mainly belong to two families *Ruminococcaceae* and *Lachnospiraceae*. The genus *Anaerostipes* is also among the 15 most abundant taxa and butyrogenic groups in a healthy microbiome[20] and its species are among the most efficient lactate consumers in the human colon[21]. In addition, it has been reported that the abundance of *Anaerostipes* was significantly decreased in type 2 diabetes subjects from African and European origin[22,23]. Several studies have shown that bacteria of this genus are capable of interacting with other microbes with diverse catabolic capacities that produce lactate[24–26]. There are presently four known *Anaerostipes* species, including *A. caccae*, *A. rhamnosivorans*, *A. hadrus* and *A. butyraticus*[27–30]. The species of this genus are not only able to use a broad range of carbohydrates but also lactate plus acetate for butyrogenesis. Whereas *A. hadrus* can use only D-lactate, *A. caccae* and *A. rhamnosivorans* are able to use both the D- and L-lactate stereoisomers, that are widely produced in the human intestinal tract and accumulated in patients with gastrointestinal disorders[31]. *A. rhamnosivorans* is among the most versatile species within the genus and the only one having the capability of fermenting dietary rhamnose to butyrate.

Inositol is a six-carbon cyclitol that is naturally present in a variety of stereoisomers. The best studied isomer is *myo*-inositol that is most abundantly present in the brain and other mammalian tissues[32] as well as many foods and fruits in particular[33]. *Myo*-inositol is involved in a large number of cellular functions, including development and function of peripheral nerves, osteogenesis and reproduction[34]. In recent years, several studies have suggested that orally administered *myo*-inositol and its isomer *chiro*-inositol can facilitate insulin signalling, improve insulin sensitivity and reduce blood glucose concentration in human disorders associated with insulin resistance, including type 2 diabetes, polycystic ovary syndrome and gestational diabetes[34–37]. It has been proposed that *myo*-inositol functions as an insulin-sensitising substance by acting on insulin receptors so that insulin can bind effectively, thus reducing insulin resistance[34]. However, the exact mechanism by which *myo*-inositol or other inositol stereoisomers exert potential beneficial effects is not known. Intriguingly, *myo*-inositol can be released from dietary phytate by bacterial phytase activity in the colon[38]. It has been shown that phytate supplementation in a rat model increased propionate production in the stool after 3-week intervention[39] and a phytate-derived metabolite from the microbial conversion has shown to promote HDAC3 activity in the gut[40]. In addition, phytate is abundantly present the vegetarian diet that has been reported to increase among others the abundance of *Anaerostipes* spp.[41]. Till date, only a few bacteria have been shown to have capabilities of metabolising *myo*-inositol. *Myo*-inositol fermentation was first reported in *Aerobacter aerogenes*, producing ethanol, $CO_2$, acetic acid and succinic acid as major end products[42] and the degradation pathway was reported via the *keto*-inositol[43]. More recently, *Bacillus subtilis* was found to convert *myo*-inositol to dihydroxyacetone phosphate, acetyl-CoA and $CO_2$ as metabolites from inositol[44] while *Lactobacillus casei* BL23 only metabolised *myo*-inositol under aerobic conditions to produce dihydroxyacetone-phosphate, acetaldehyde, acetyl-CoA and $CO_2$[45]. The *myo*-inositol cascade till to dihydroxyacetone has been described previously and the enzymes involved in this part of the pathway have been biochemically and genetically characterised in *Lactobacillus casei*[45] and *Bacillus subtilis*, where these are annotated as products of the *iolABCDEFGHIJ* operon[44]. However, these bacteria all degrade *myo*-inositol under aerobic conditions and are not naturally found in the human intestine. In this study we report a pathway involved in the conversion of *myo*-inositol or *chiro*-inositol to propionate and acetate that was present in many but not all tested strains of *Anaerostipes* spp. This unique pathway was further elucidated using [13]C-labelled inositol isotopomers and nuclear magnetic resonance (NMR), mass spectrometry and proteogenomic analyses in *A. rhamnosivorans* revealing a set of enzymes involved in intestinal propionate formation that were only found to be encoded in the genomes of *Anaerostipes* and related species. Metagenomic analysis of the gut microbiota in prediabetes/diabetes cohorts showed a significant correlation of the propionate pathway genes with biomarkers of insulin sensitivity. This was in line with the observed reduction of fasting glucose in mice fed a western diet and treated with live *A. rhamnosivorans* in the presence of *myo*-inositol. Altogether this study provides the molecular details of the conversion of dietary inositol stereoisomers into propionate and acetate by an abundant human intestinal commensal, which is part of a phytate food chain in the gut and further studies in human are needed to investigate the potential use of *Anaerostipes* spp. to improve host health.

## Results

**Inositol fermentation by *Anaerostipes* species**. *Anaerostipes* is most known for its capability of producing butyrate from sugars as well as lactate and acetate[46]. Interestingly, we found several *Anaerostipes* strains to be able to utilise *myo*-inositol to produce mainly equal amounts of propionate and acetate and minor amounts of formate, hydrogen, lactate and butyrate (Fig. 1b) while in absence of inositol no products were detected by any of the strains (Supplementary Fig. 1). Both *A. rhamnosivorans* DSM26241[T] and *A. caccae* DSM14662[T] metabolised inositol rapidly (both with doubling times of 1.7 h) while strain *A. hadrus* PEL85 showed some adaptation and grew somewhat slower (doubling time 3.4 h) (Fig. 1a). All three *Anaerostipes* strains completely consumed 15 mM *myo*-inositol after 33 h (Fig. 1b). In contrast, *A. hadrus* DSM3319[T], *A. hadrus* DSM108065 and *A. butyraticus* DSM22094[T] did not ferment *myo*-inositol, even not

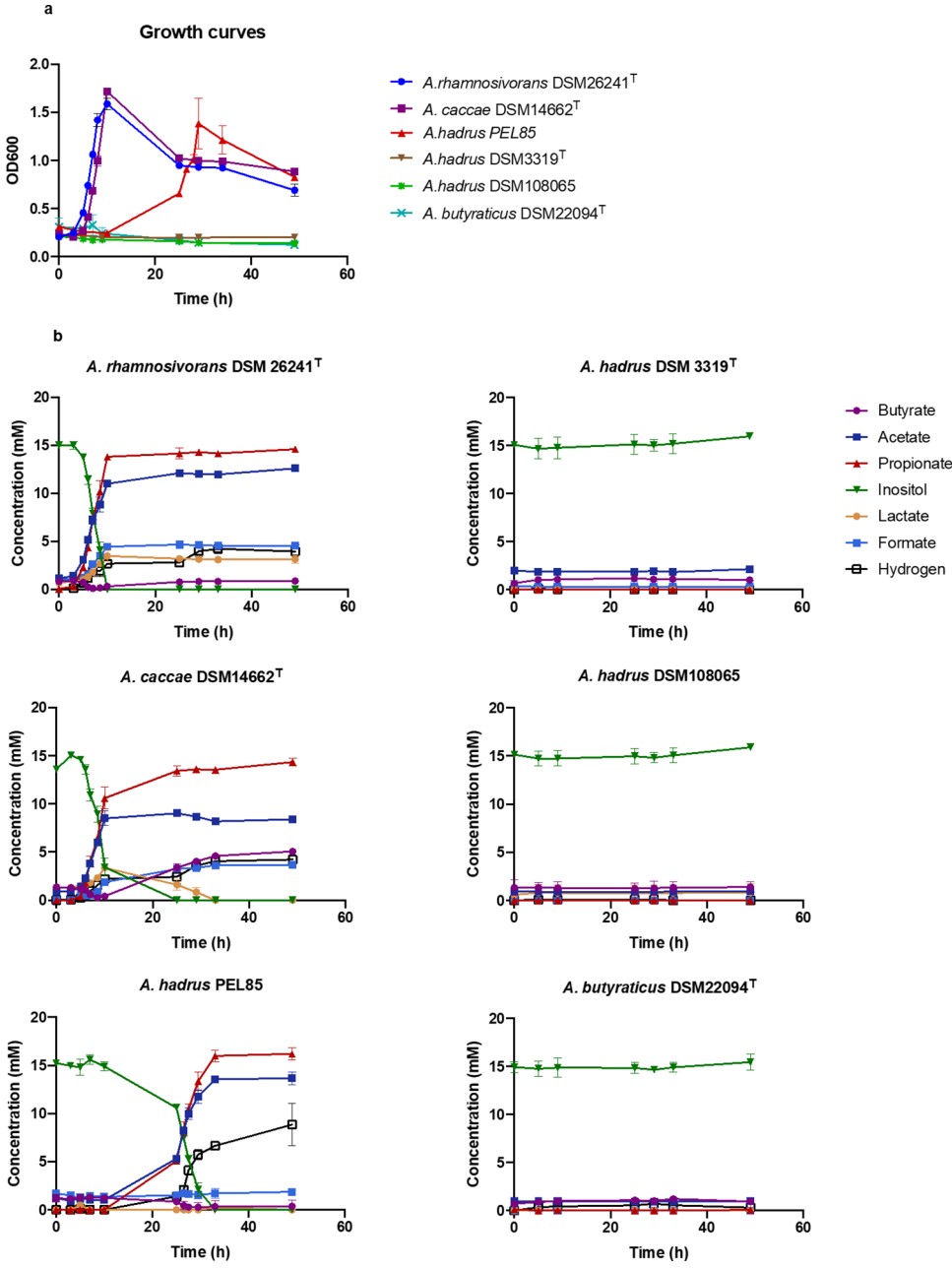

**Fig. 1 Metabolite production from *myo*-inositol by *Anaerostipes* strains.** The growth was performed in bicarbonate-buffered medium supplemented up to 15 mM *myo*-inositol as substrate. **a** Growth curves ($n = 3$ biological replicates); **b** Metabolite production and substrate consumption by 6 *Anaerostipes* strains ($n = 3$ biological replicates). Data are presented as mean values ±SD.

after prolonged incubation times. A small amount of butyrate (<1.5 mM) was produced by all *myo*-inositol degraders and non-degraders, suggesting that this was likely a fermentation product of the yeast extract and peptone present in the medium or a carry-over carbon source from the inoculum. Of note, *A. caccae* DSM14662[T] first produced lactate and subsequently metabolised this and simultaneously produced butyrate, indicating that butyrate formation was mostly from lactate (Fig. 1b). *A. rhamnosivorans* DSM26241[T] and *A. hadrus* PEL85 performed a similar conversion inside the cells that resulted in little accumulation of lactate during growth (Fig. 1). Based on the observed stoichiometry of *myo*-inositol conversion to propionate and acetate, we propose the theory fermentative equation as: $2C_6H_{12}O_6 \rightarrow 2C_3H_6O_2 + 2C_2H_4O_2 + CO_2 + H_2 + CH_2O_2$. Of note, the production of $CO_2$ was not determined due to the pre-existing

bicarbonate in the medium, however $CO_2$ was clearly detected by NMR analysis in the following section. In addition, we did not observe any influence of additional acetate on propionate formation (Supplementary Fig. 2).

**Elucidation of *myo*-inositol fermentation pathway using [13]C-NMR and mass spectrometry.** To elucidate the *myo*-inositol pathway at the molecular level we used *A. rhamnosivorans* DSM26241[T] as a model because of its rapid growth and used [13]C-labelled substrates in whole-cell NMR analysis. *A. rhamnosivorans* was grown in bicarbonate-buffered medium containing either [$^{13}C_6$]*myo*-inositol, [4-$^{13}$C]*myo*-inositol or [4,5-$^{13}C_2$]*myo*-inositol as the energy and carbon source. The supernatants were collected during the growth and analysed by NMR. Growth of *A.*

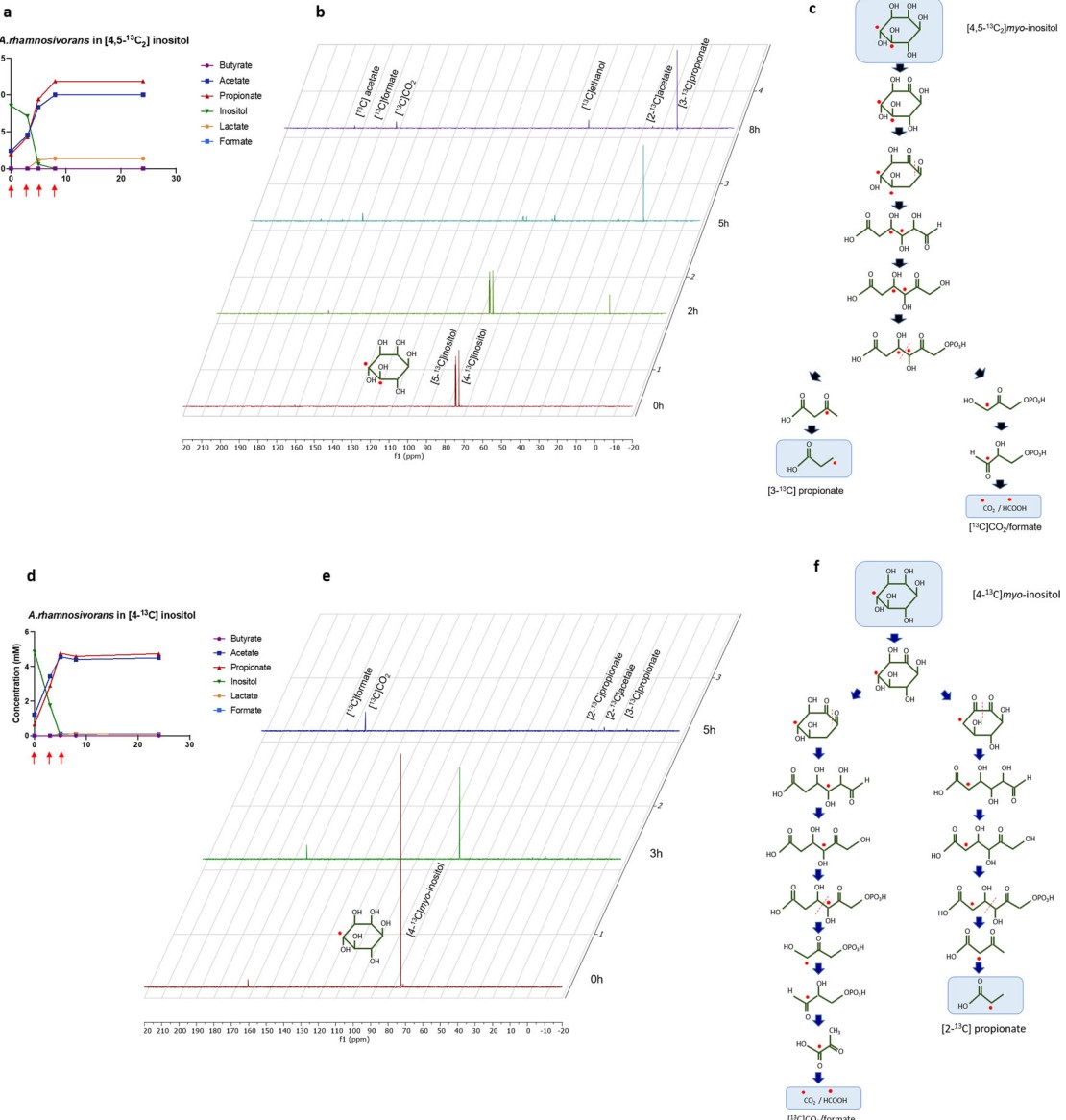

**Fig. 2 Elucidation of *myo*-inositol pathway via $^{13}$C-NMR analysis.** High-resolution $^{13}$C-NMR spectra showing [4,5$^{13}$C$_2$]*myo*-inositol fermentation products that are [3-$^{13}$C]propionate, [$^{13}$C]CO$_2$, [1-$^{13}$C]ethanol, [2-$^{13}$C]acetate, [1-$^{13}$C]acetate, [$^{13}$C]formate (**b**) with anticipated scheme of $^{13}$C flow (**c**), and [4-$^{13}$C]*myo*-inositol fermentation products that are [$^{13}$C]CO$_2$, [$^{13}$C]formate, [2-$^{13}$C]propionate, [2-$^{13}$C]acetate and [3-$^{13}$C]propionate (**e**) with anticipated scheme of $^{13}$C flow (**f**). **a** and **d** show concentrations of substrate and end metabolites analysed using high performance liquid chromatography-refractive index detection (HPLC-RI) when the cells were grown in [4,5$^{13}$C$_2$]*myo*-inositol and [4-$^{13}$C]*myo*-inositol respectively. Red arrows indicate the samples time points for $^{13}$C-NMR.

*rhamnosivorans* on [4,5-$^{13}$C$_2$]*myo*-inositol resulted in its complete conversion into [3-$^{13}$C]propionate as major end product and minor amounts of [$^{13}$C]CO$_2$, [1-$^{13}$C]ethanol, [2-$^{13}$C]acetate, [1-$^{13}$C]acetate, [$^{13}$C]formate after 8 h incubation (Fig. 2a, b). As only single peaks were detected for all end products, there were no compounds with adjacent $^{13}$C nuclei produced from [4,5-$^{13}$C$_2$]*myo*-inositol. This indicated that the cleavage of 5-dehydro-2-deoxy-D-gluconate 6-phosphate occurred between C3 and C4, resulting in [3-$^{13}$C]propionate, [$^{13}$C]CO$_2$, and [$^{13}$C]formate as end products (Fig. 2c). The labelled ethanol and acetate might have been produced from a conversion of intermediate labelled acetyl-CoA formed in *myo*-inositol pathway (Fig. 3b). This possibility was confirmed by the results of labelled compounds detected from cells grown in [4-$^{13}$C]*myo*-inositol. We ran these samples with the number of scans of 1024 to get visible peaks (Fig. 2d, e). [$^{13}$C]CO$_2$ was found as major end product and small

quantities of [$^{13}$C]formate, [2-$^{13}$C]propionate, [2-$^{13}$C]acetate and [3-$^{13}$C]propionate. This result not only confirms the cleavage of 5-dehydro-2-deoxy-D-gluconate 6-phosphate between C3 and C4, forming [$^{13}$C]CO$_2$ and [$^{13}$C]formate but also indicates that dehydration of *scyllo*-inosose was mainly across the C4–C5 bond with a small portion of which the dehydration was across the C1–C6 bond (Fig. 2f). Importantly, no labelled butyrate was detected in any condition, confirming that butyrate is not the end product of inositol fermentation. In addition, we also grew *A. rhamnosivorans* in [$^{13}$C$_6$]*myo*-inositol, resulting in the formation of labelled compounds including [$^{13}$C$_3$]propionate, [$^{13}$C$_2$]acetate, [$^{13}$C]CO$_2$, [$^{13}$C]formate, [$^{13}$C$_3$]lactate, [$^{13}$C$_2$]ethanol and [2-$^{13}$C]butyrate (Supplementary Fig. 3). While all other products were labelled in all carbon atoms, only one labelled carbon was detected at C2 position of butyrate. The fact that labelled lactate was formed along with labelled butyrate, supported the

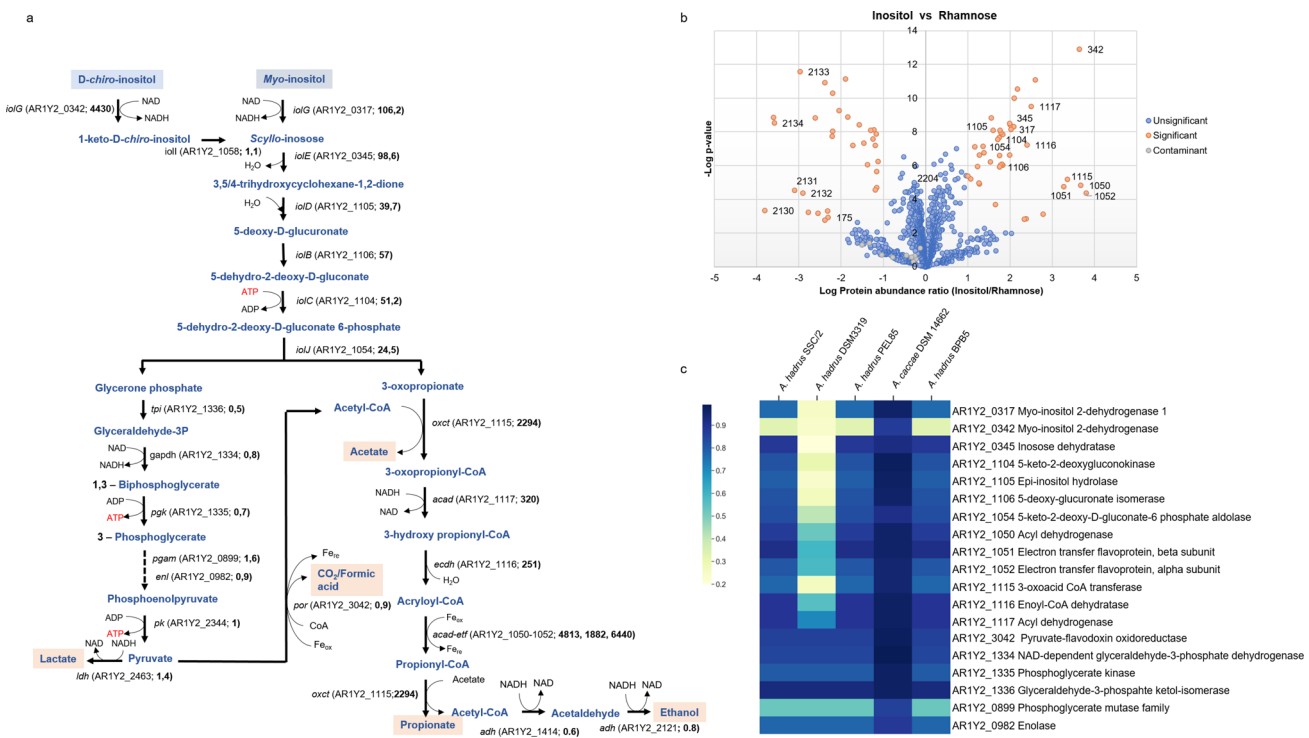

**Fig. 3 Postulated myo-inositol metabolic pathway via proteogenomic analysis. a** Differential protein abundance in *myo*-inositol and rhamnose. The T-test result shows that proteins are considered to be significantly (orange dots) different between the two conditions when there is a difference of a factor 10 or larger between the Inositol and Rhamnose condition (that is Log10 Protein abundance ratio (Inositol/Rhamnose) below −1 or above 1) and a *p*-value below 0.002 (−Log10 *p*-value > 2.7). Contaminants (human keratins and bovine trypsin), shown as grey dots, show poor *p*-values as expected since they should not be significantly different (-Log10 *p*-values below 1.4 = *p* > 0.04). Other proteins identified and quantified do not show a significantly different abundance (blue circles) between the Inositol and Rhamnose condition. **b** *myo*-inositol metabolic pathway. The locus tag and fold induction of the proteins (bold) based on the proteomic data are indicated for each reaction. *Myo*-inositol is first metabolised via a step-wise reaction by *myo*-inositol dehydrogenase (IolG encoded by AR1Y2_0317), inosose dehydratase (IolE encoded by AR1Y2_0345), epi-inositol hydrolase (IolD encoded by AR1Y2_1105), 5-deoxy-glucuronate isomerase (IolB encoded byAR1Y2_1106) and 5-keto-2-deoxygluconokinase (IolC encoded by AR1Y2_1104) before being cleaved off to glycerone phosphate and 3-oxopropionate by 5-keto-2-deoxy-ᴅ-gluconate-6 phosphate aldolase (IolJ encoded by AR1Y2_1054). While glycerone phosphate is further converted to pyruvate via glycolysis, 3-oxopropionate enters a reduction branch involved newly identified enzymes including a 3-oxoacid CoA transferase (OxcT encoded by AR1Y2_1115), an enoyl-CoA dehydratase (AcaD encoded by AR1Y2_1116) and an acyl dehydrogenase (EcdH encoded by AR1Y2_1117) and acyl dehydrogenase complex (AcaD-Etf encoded by AR1Y2_1050-1052). $CO_2/H_2$ or formate is also formed from a conversion of pyruvate to acetyl-CoA involved pyruvate-flavodoxin oxidoreductase (Por encoded by AR1Y2_3042). Genes involved in a production of side products (lactate and ethanol) are also indicated in the graph. **c** Homologues of *myo*-inositol pathway gene candidates in genomes of *Anaerostipes* isolates. The similarity of each pathway gene to this of *A. rhamnosivorans* is indicated via colour code from yellow to blue ranging from low to high. The genes and locus tags of the myo-inositol pathway genes in the genome of *A. rhamnosivorans* are shown on the right of the figure.

explanation that the small amount of butyrate observed in the growth experiments (Fig. 1) was derived from lactate/acetate rather than *myo*-inositol. This proposed route is in line with the detection of intermediates by liquid chromatography high-resolution mass spectrometry (LC-MS/MS). We detected 4 key intermediates of this pathway, including *scyllo*-inosose; 3,5/4-trihydroxycyclohexane-1,2-dione; 5-dehydro-2-deoxy-ᴅ-gluconate and 5-dehydro-2-deoxy-ᴅ-gluconate 6-phosphate (Supplementary Fig. 4 and Supplementary Table 4). While the first three intermediates were accumulated rapidly and subsequently utilised, 5-dehydro-2-deoxy-ᴅ-gluconate 6-phosphate was only detected in small quantities in the first 2 h and quickly consumed. This rapid consumption may explain the inability to detect the following intermediates, including 3-oxopropionyl-CoA and 3-oxopropionate. Another reason might be due to their short half-times that limit the mass spectrometry analysis. Based on the NMR and mass spectrometry data, the entire metabolic *myo*-inositol pathway was reconstructed (Fig. 3a) and further investigated by proteogenomic approach in combination with enzyme analysis (see below).

**Proteogenomic and comparative genomic analyses revealed the presence of entire *myo*-inositol pathway to propionate in *Anaerostipes* spp.** The complete genome of *A. rhamnosivorans* DSM26241[T] was determined by single molecule sequencing and found to contain 3,588,860 bp with a GC content of 44.5% coding for a total of 350 subsystems with 3556 protein-coding sequences (CDSs) (NCBI accession number CP040058). Complete acetyl-CoA and rhamnose degradation pathways were detected and predicted to operate via butyryl-CoA:acetate CoA transferase (EC 2.8.3.8), the enzyme that is involved in the final step of the butyrate synthesis pathway (Supplementary Table 2). Lactate permease and lactate dehydrogenase were also found encoded in the genome, which explains the capability of *A. rhamnosivorans* to use lactate[28]. In addition, gene candidates for *myo*-inositol metabolic pathway were identified in the annotated genome of *A. rhamnosivorans* DSM26241 (Fig. 3a). *Myo*-inositol catabolism involved multiple and stepwise reactions including 6 genes (*iolBCDEGJ*) for the conversion of *myo*-inositol to glycerone phosphate and 3-oxopropionate[44] (Fig. 3A). Notably, 11 copies of myo-inositol 2-dehydrogenase (*iolG*) and 4 copies of inosose

dehydratase (*iolE*) were found in the genome, indicating the adaptation of *A. rhamnosirovans* to this cyclic polyol. Energy is predicted to be conserved via substrate level phosphorylation as well as electron transport chain by acryloyl-CoA dehydrogenase/ Etf complex (AR1Y2_1050-1052)[47]. Several permeases were also found in the genome with a putative transporter gene (AR1Y2_1059) located closed to *myo*-inositol catabolic genes (Supplementation Fig. 5). To elucidate proteins involved in the propionate production from 3-oxopropionate, we analysed proteins extracted from A. *rhamnosivorans* DSM26241[T] grown on either *myo*-inositol or rhamnose by semi-quantitative proteomics using mass spectrometry (Supplementary Tables 1–2 and Supplementary data 2). The proteins predicted to be involved in *myo*-inositol and rhamnose fermentation were highly abundant in the respective proteomes (Fig. 3b). We found under rhamnose condition, *A. rhamnosivorans* DSM26241[T] produced abundantly all proteins involved in rhamnose fermentation and employed a rhamnose pathway[48] and acetyl-CoA pathway[49], forming 1,2-propanediol and butyrate as final metabolites. Interestingly, all proteins involved in *myo*-inositol pathway were not only highly abundant but also significantly induced during growth on *myo*-inositol compared with rhamnose (Fig. 3b). Enzymes involved in the six stepwise conversions of *myo*-inositol to glycerone phosphate and 3-oxopropionate were increased up to 106-fold in the proteome of *myo*-inositol-grown cells, including *myo*-inositol 2-dehydrogenase (IolG encoded by AR1Y2_0317), inosose dehydratase (IolE encoded by AR1Y2_0345), *epi*-inositol hydrolase (IolD encoded by AR1Y2_1105), 5-deoxy-glucuronate isomerase (*IolB* encoded by AR1Y2_1106), 5-keto-2-deoxygluconokinase (IolC encoded by AR1Y2_1104) and 5-keto-2-deoxy-D-gluconate-6-phosphate aldolase (IolJ encoded by AR1Y2_1054). Among these, *iolCDB* (AR1Y2_1104-1106) were located in a putative operon and they shared similar induced protein abundances in cells grown on *myo*-inositol. A transcriptional repressor (IolR encoded by AR1Y2_1091) was also found a few genes downstream from the *iolCDB* operon (Supplementary Table 1 and Supplementary Fig. 5). Remarkably, 2 other myo-inositol 2-dehydrogenases (IolG encoded by AR1Y2_0342 and AR1Y2_1057) were abundantly produced in the *myo*-inositol condition along with a predicted inosose isomerase (IolJ encoded by AR1Y2_1058) (Supplementary Table 1). As the *myo*-inositol 2-dehydrogenase and the inosose isomerase have been reported to confer the ability to convert *chiro*-inositol to *scyllo*-inosose, an intermediate of *myo*-inositol fermentation[44], the over-production of these enzymes by *A. rhamnosivorans* indicates the capability of this strain to metabolise *chiro*-inositol, an isomer of *myo*-inositol as well. We also confirmed this ability of *chiro*-inositol conversion experimentally (Supplementary Fig. 6). The conversions from glycerone phosphate to acetyl-CoA were predicted as shared steps between *myo*-inositol and rhamnose pathway. Hence, no remarkable differences were found for the abundance of these proteins in cells grown on either rhamnose or *myo*-inositol. In contrast, enzymes involved in the conversion of 3-oxopropionate to propionate were most abundant and substantially induced, up to 6440 times, during growth on *myo*-inositol compared to rhamnose. These included 3-oxoacid CoA transferase (OxcT encoded by AR1Y2_1115), acyl-CoA dehydrogenase (AcaD encoded by AR1Y2_1117), enoyl-CoA hydratase (EcdH encoded by AR1Y2_1116) and acryloyl-CoA dehydrogenase/ Etf (AcaD-Etf encoded by AR1Y2_1050-1052). Interestingly, *oxcT, ecdH* and *acaD* (AR1Y2_1115-1117) were located in a putative operon. The 3-oxoacid CoA transferase protein (OxcT encoded by AR1Y2_1115) was predicted to be involved in two key reactions, one converting 3-oxopropionate to 3-

oxopropionyl-CoA and the other converting propionyl-CoA to propionate (Fig. 3a), that explained the high (2294-fold) induction in *myo*-inositol-grown cells (Fig. 3b). The substantial increase in the abundance of 3-oxoacid CoA transferase in cells grown on *myo*-inositol could partially explain the absence of accumulation of either 3-oxopropionyl-CoA or 3-oxopropionate, as we were not able to detect these intermediates by reversed phase LC-MS/MS (see above; the Methods section and Supplementary Fig. 4). The acryloyl-CoA dehydrogenase/Etf complex (AcaD-Etf encoded by AR1Y2_1050-1052) was predicted to be involved in energy conservation; hence its abundance was much higher in *myo*-inositol grown as compared to rhamnose-grown cells (up to 6440-fold). It is predicted that the production of small amounts of ethanol and lactate (see above) was likely needed to prevent the excess reducing equivalence produced during *myo*-inositol fermentation. Since both ethanol and lactate production also took place during rhamnose metabolism, the proteins involved in these reactions were not induced in abundance and were not considered further.

To investigate how widely spread this newly discovered pathway is in *Anaerostipes* spp., we compared the *A. rhamnosivorans* genome with that of the publicly available genomes of *A. caccae, A. hadrus* SSC/2, *A. hadrus* PEL85, *A. hadrus* DSM3319[T] and *A. hadrus* BPB5. The genomes of the four *A. hadrus* strains were highly similar with more than 97% similarity for average nucleotide identity (ANI) and substantially different from genomes of *A. rhamnosivorans* and *A. caccae* with only 70% ANI similarity (Supplementary Table 3). Homologues of *myo*-inositol pathway genes were determined by blasting amino acid sequences of gene candidates from *A. rhamnosivorans* on other genomes (Fig. 3c). Intriguingly, most strains have homologues of *myo*-inositol pathway genes except *A. hadrus* DSM3319[T], indicating the adaptation of *Anaerostipes* strains to substrate availability in the intestine. These results are in line with the growth tests in which *A. rhamnosivorans, A. caccae* and *A. hadrus* PEL85 were able to convert *myo*-inositol to propionate while *A. hadrus* DSM3319[T] was not (Fig. 1). Detailed analysis of the *myo*-inositol pathway gene localisation showed that this pathway composed of several gene clusters that were shared by all *Anaerostipes* strains. The arrangement of these gene clusters was almost identical among *A. hadrus* strains while that of *A. rhamnosivorans* was highly similar to that of *A. caccae* (Supplementary Fig. 5). Furthermore, a homologue of *myo*-inositol 2-dehydrogenase (IolG) (encoded by AR1Y2_0342) was only found in *A. caccae* but absent in all other strains while homologues of the other *myo*-inositol 2-dehydrogenase (encoded by AR1Y2_1057) was present in all strains. To investigate which IolG homologue was involved in *chiro*-inositol conversion, we incubated *A. rhamnosivorans, A. caccae, A. hadrus* DSM3319[T] and *A. hadrus* PEL85 in *chiro*-inositol. Intriguingly, only *A. rhamnosivorans* and *A. caccae*, which had the homologue of AR1Y2_0342, were able to convert *chiro*-inositol to propionate and acetate while *A. hadrus* DSM3319[T] and *A. hadrus* PEL85 were not (Supplementary Fig. 6). These results suggest *myo*-inositol 2-dehydrogenase (IolG encoded by AR1Y2_0342) likely conferred the ability of metabolising *chiro*-inositol.

**A specific CoA transferase involved in propionate production.** Enzymes from CoA transferase family are known to be involved in propionate and butyrate formation from many different substrates, including sugars and amino acids[49–51]. A total of nine CoA transferases were found to be encoded by the genome of *A. rhamnosivorans* but only two of these were detected in the proteomic analysis with high abundance and included 3-oxoacid

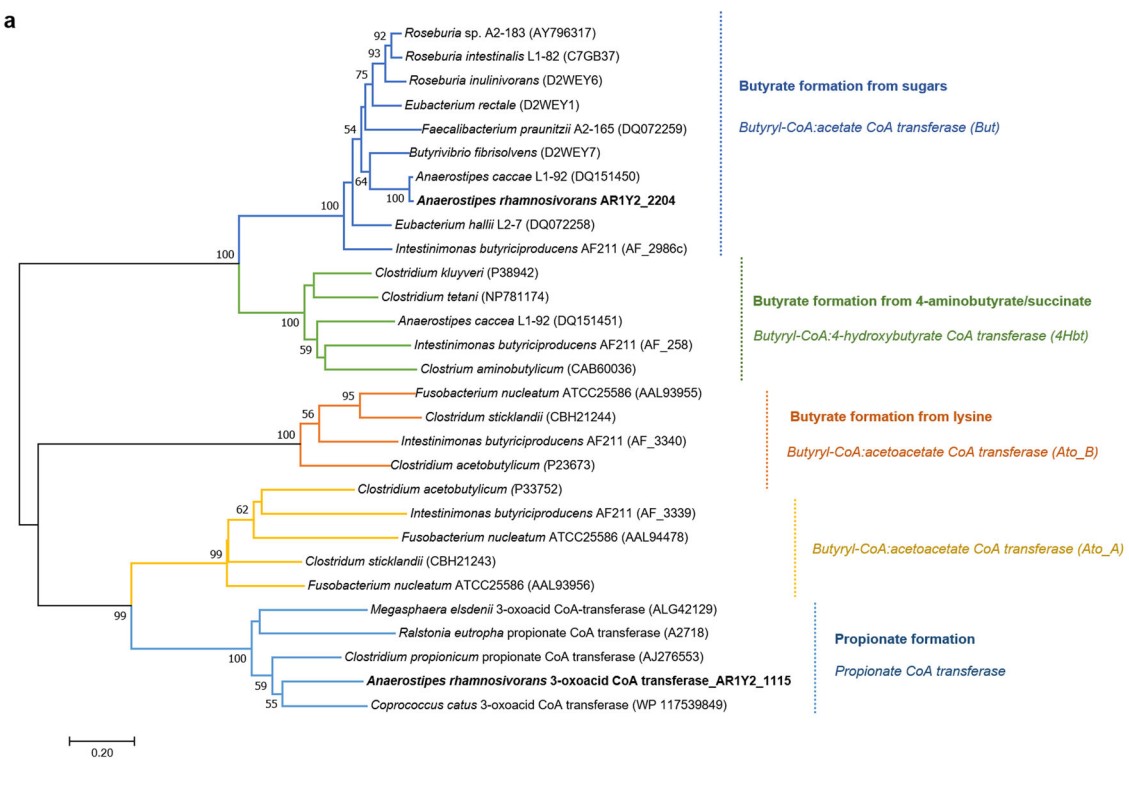

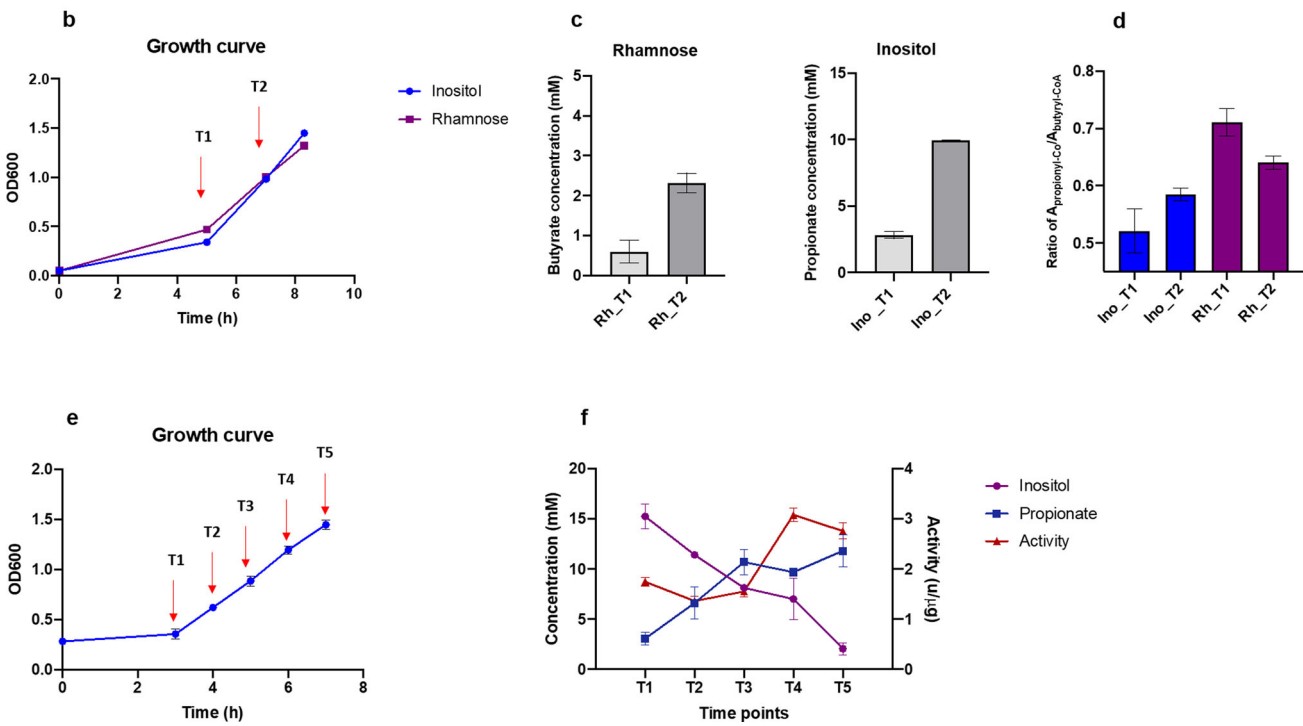

CoA transferase encoded by AR1Y2_1115 and butyryl-CoA: acetate CoA transferase encoded by AR1Y2_2204 (Fig. 3b and Supplementary Tables 1–2). These CoA transferases and canonical CoA transferases from other anaerobes were used to construct a phylogenetic tree, forming five distinct clades corresponding to five different reactions (Fig. 4a). Butyryl-CoA: acetate CoA transferase, butyryl-CoA:4-hydroxybutyrate CoA transferase, butyryl-CoA:acetoacetate CoA transferase and propionate CoA transferase are involved in butyrate formation from

sugars, 4-aminobutyrate/succinate, lysine and propionate formation respectively. While the protein with locus tag AR1Y2_2204 was clustered with butyryl-CoA:acetate CoA transferase from butyrate-producing bacteria, that with locus tag AR1Y2_1115 was found to cluster with well-studied propionate CoA transferases of other anaerobic bacteria, including the 3-oxoacid/propionate CoA transferases from *Coprococcus catus* (WP117539849); *Megasphaera elsdenii* (ALG42129); *Ralstonia eutropha* H16 (A2718) and *Clostridium propionicum* (AJ276553)[13,52–54]. In

**Fig. 4 Identification of propionyl coA transferase in *A. rhamnosivorans*.** Phylogeny of CoA transferase family (**a**) and CoA transferase activity (**b**–**f**). **a** Phylogenetic tree of predicted CoA transferases from *A rhamnosivorans* DSM26241[T] (bold) and other anaerobes. The tree was constructed using sequences from butyryl-CoA:acetate CoA transferase (But, in blue), butyryl-CoA:4-hydroxybutyrate CoA transferase (4Hbt, in green), butyryl-CoA: acetoacetate CoA transferase (Ato) alpha subunit (AtoD, in orange), beta subunit (AtoA, in yellow) and oxoacid CoA transferase (OxcT, in light blue), respectively. **b** Growth curves of *A. rhamnosivorans* on *myo*-inositol and rhamnose of which cells at 2 time points were harvested for crude extracts and enzyme assays. **c** Butyrate and propionate production from rhamnose and *myo*-inositol condition at T1 and T2 (*n* = 2 technical replicates). Data are presented as mean values ±SD. **d** Ratios between activities on propionyl-CoA out of butyryl-CoA in all anaerobic cell-free extracts with blue and purple column for crude extracts from *myo*-inositol and rhamnose respectively. CoA transferase activities for butyryl-CoA and propionyl-CoA in anaerobic cell-free cell extracts were determined and subsequently used for ratio calculation. Each measurement was performed in triplicate (*n* = 3 technical replicates). Data are presented as mean values ±SD. Cell free extract of *Intestinimonas butyriciproducens* AF211 grown on lysine was used as a negative control (NC). As strain AF211 did not show any activities toward either butyryl-CoA or propionyl-CoA, the ratio result of this strain was omitted from the graph to avoid confusion. **e** Growth curve of *A. rhamnosivorans* in *myo*-inositol (*n* = 2 biological replicates). Red arrows show sampling points (T1–T5) of which enzyme extracts were obtained for the assay. Data are presented as mean values ±SD. **f** Increases in enzyme activities toward propionyl-CoA during exponential phase correspond to an increase in propionate production and inositol consumption. Data of CoA activity are presented as mean values of biological duplicates (*n* = 2) and technical triplicates (*n* = 3) while values shown for inositol and propionate concentration are means of biological duplicates (*n* = 2). Data are presented as mean values ±SD.

addition, the abundance of the protein with locus tag AR1Y2_2204 was only 1.7 times higher in cells grown on rhamnose as compared to *myo*-inositol, suggesting this protein is constitutively expressed. Remarkably, the protein with locus tag AR1Y2_1115 was 2294-times more abundant in *myo*-inositol than rhamnose-grown cells, indicating that 3-oxoacid CoA transferase (OxcT encoded by AR1Y2_1115) functions as propionate CoA transferase and is actively involved in propionate formation from *myo*-inositol.

To further confirm this propionate CoA transferases activity, anaerobic cell-free extracts were prepared from cells grown on *myo*-inositol and rhamnose. Subsequently, we performed anaerobic activity assays using propionyl-CoA or butyryl-CoA and acetate as substrates (Fig. 4b–d). The activity was measured indirectly via a coupling reaction with citrate synthase assay[55]. While *A. rhamnosivorans* only produced propionate (and acetate) from *myo*-inositol, butyrate was detected from rhamnose (Fig. 4c), suggesting that two different CoA transferases were involved in propionate and butyrate formation. The activity for propionyl-CoA and butyryl-CoA was found in all anaerobic cell-free extracts during the exponential phase while no activity was found in a negative control (an enzyme extract of *Intestinimonas butyriciproducens* grown on lysine and known to contain butyryl-CoA: acetoacetate CoA transferase[50]). This is not surprising as CoA transferases have been reported to have a broad substrate specificity[56]. Specifically, the CoA transferase activities of the anaerobic cell-free extracts from rhamnose-grown cells were higher than these from *myo*-inositol-grown cells regardless of substrates and growth phase, indicating two different CoA transferases being active in two growth conditions. Additionally, during exponential phase the ratio of CoA transferase activity for propionyl-CoA over that of butyryl-CoA (0.52 up to 0.57) was increased in cells grown on *myo*-inositol but decreased in those grown on rhamnose (0.71 down to 0.64) (see Fig. 4d). Furthermore, the induction of the 3-oxoacid CoA transferase encoded by AR1Y2_1115 in the inositol condition was in agreement with the increased propionate CoA transferase activities along with propionate production from inositol during exponential phase with (Fig. 4e, f). These results indicate an active role of 3-oxoacid CoA transferase (OxcT encoded by AR1Y2_1115) in propionate production from *myo*-inositol while butyryl-CoA:acetate CoA transferase (But encoded by AR1Y2_2204) was involved in butyrate formation from rhamnose.

**Metagenomic analysis and faecal enrichment on [¹³C₆]phytate indicate the physiological relevance of inositol fermentation by *Anaerostipes* in human gut.** To further investigate the relevance of *myo*-inositol metabolism in the intestinal tract, we analysed metagenomes of 65 subjects from a healthy population characterised in the Human Microbiome Project[1]. The results revealed the presence of the *myo*-inositol pathway genes in the vast majority (>80%) of the cohort with a relative abundance of ~0.3% of the annotated genes (Supplementary Fig. 7). In addition, genes coding for phytases derived from *Bifidobacterium* spp. were also detected in a quarter of the entire population, notably *B. longum* subsp. *infantis* and *B. pseudocatenulatum*. The co-occurrence of the *myo*-inositol pathway and phytase genes in these subjects indicated the presence of a phytate food chain involving *Anaerostipes* spp. in the intestinal tract. The functionality of such a trophic chain was demonstrated by coculturing *A. rhamnosivorans* and *Bifidobacterium longum* subsp. *infantis* on a medium containing phytate (Supplementary Fig. 8). This resulted in the accumulation of a small but persistent amount *myo*-inositol in a monoculture of *B. longum* subsp. *infantis* and production of lactate and acetate from the medium. In contrast, in the presence of both *B. longum* subsp. *infantis* and *A. rhamnosivorans*, no accumulation of *myo*-inositol was observed and instead greater propionate formation observed. Although butyrate was detected in the coculture of *B. longum* subsp. *infantis* and *A. rhamnosivorans*, a similar amount of butyrate was already produced by *A. rhamnosivorans* when grown in a medium without phytate (Supplementary Fig. 8). This implies the butyrate formation was likely from the fermentation of medium components by *A. rhamnosivorans* rather than phytate. These results indicate the cross-feeding between *B. longum* subsp. *infantis* and *A. rhamnosivorans* on phytate via *myo*-inositol interspecies-transfer, leading to an increased propionate production.

To further support the presence of a phytate food chain in the intestinal tract leading to propionate, we performed enrichment studies with [¹³C₆] phytate using faecal samples from two healthy donors with low levels of endogenous *Anaerostipes* species (Supplementary Fig. 9) and analysed the end metabolites using [¹³C] NMR (see Fig. 5 and Supplementary Fig. 11). Our results showed that microbiota of these two donors converted [¹³C₆] phytate into [¹³C]acetate and [¹³C]butyrate as end metabolites (Fig. 5a). *A. rhamnosivorans* alone was not able to metabolise [¹³C₆]phytate (Supplementary Fig. 10). However, we found that [¹³C]propionate was produced from [¹³C₆]phytate once *A. rhamnosivorans* cells were simultaneously added to the medium with the faecal inoculum (Fig. 5b–d). The SCFA analysis showed that a significant amount of propionate (together with acetate and butyrate due to the activity of faecal microbiota) was produced from the added phytate (Fig. 5c). These results imply that *A. rhamnosivorans* metabolised the inositol released from phytate

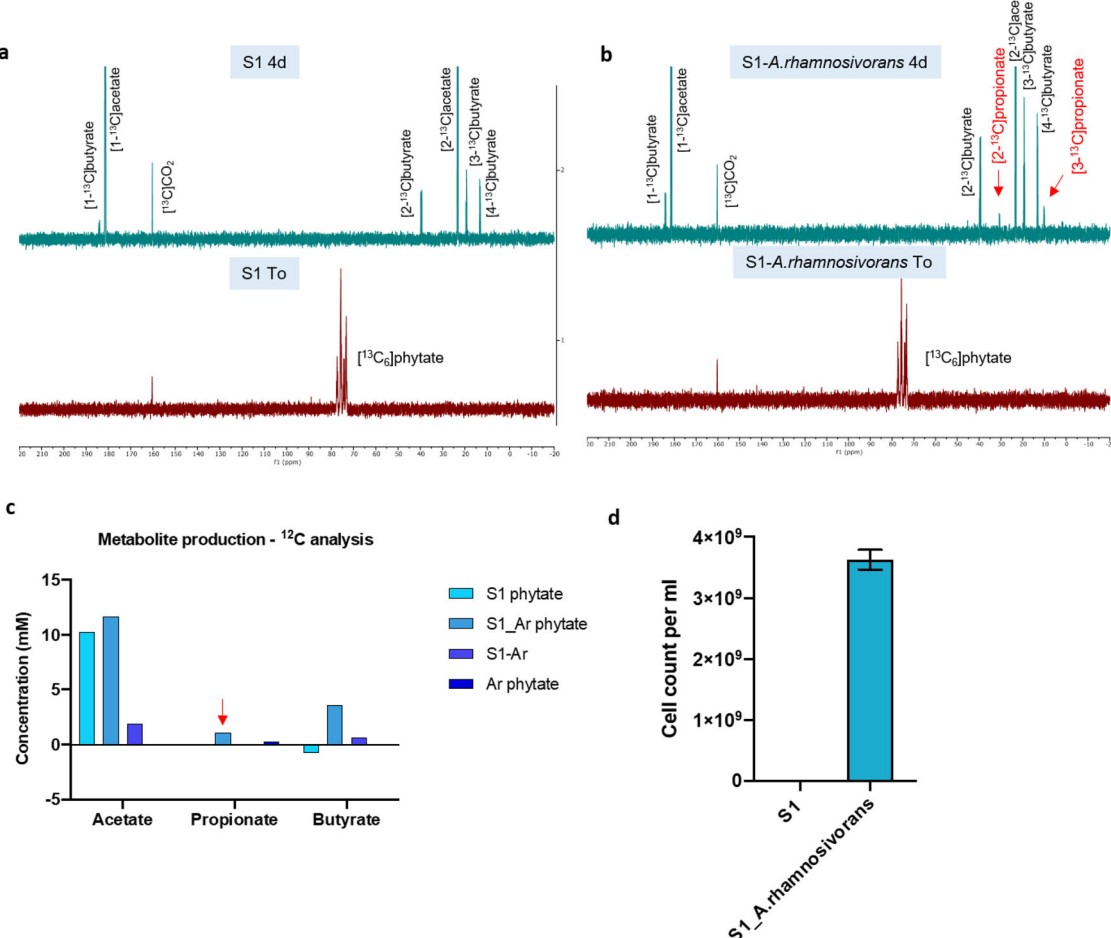

**Fig. 5 [$^{13}C_6$]-labelled phytate degradation by human microbiota in presence or absence of *A. rhamnosivorans*. a** $^{13}$C compounds detected by NMR at To or 4 day incubation in phytate enrichment using faecal microbiota. **b** $^{13}$C compounds detected by NMR at To or 4 day incubation in phytate enrichment using faecal microbiota and *A. rhamnosivorans*. **c** Quantity of metabolite production analysed by high performance liquid chromatography - refractive index detection (HPLC-RI). **d** *A. rhamnosivorans* qPCR-based cell counts at initial time points of S1 and S1 plus *A. rhamnosivorans* enrichments. Mean values of technical triplicates are shown with ±SD.

degradation by gut bacteria for propionate formation, hence forming a phytate-dependent food chain. The stable isotope analysis of the end products clearly shows the active involvement of *A. rhamnosivorans* in the phytate degradation. Similar results of phytate enrichment were obtained with the second faecal donor (Supplementary Fig. 11), confirming the role of *A. rhamnosivorans* in propionate production from phytate. Indeed, humans often consume phytate-containing plants, such as brans and cereals, and hence *myo*-inositol is likely released by microbial activities in the large intestine and serves as substrate for intestinal bacteria[57]. Our results show that dietary phytate can be utilised by gut bacteria, leading to the production of acetate, propionate and butyrate that involve different types of intestinal microbes and emphasise the key role of *A. rhamnosivorans* in propionate and acetate production from inositol or its precursor phytate.

**Anaerostipes spp. as potential health promoting bacteria.** A comparative analysis of over 10,000 available NCBI genomes showed that seven bacterial species contained the complete inositol pathway, of which five belonged to *Anaerostipes* spp., one to a bacterium belonging to the Lachnospiraceae and related to *Anaerostipes*, and one to a non-intestinal soil isolate

*Anaerosacchriphilus polymeriproducens* (Supplementary data 1). This result suggests that *Anaerostipes* is a key taxonomic group, members of which are predicted to contain the entire metabolic pathway to convert inositol to propionate.

Recently, a comprehensive metagenomic analysis of the intestinal microbiota in a large prediabetes/diabetes cohort was reported[58]. Using this dataset we performed a deep analysis of the genes involved in the inositol-to-propionate pathway and observed a significant negative association of the propionate production gene cluster but not the inositol utilisation cluster to metabolic disease biomarkers (Fig. 6a). It has been known that several bacteria are able to metabolise *myo*-inositol to acetate and ethanol employing the inositol utilisation genes[44,45] while the propionate production gene cluster has been only observed in *Anaerostipes* spp. Of interest, the relative abundance of the propionate production gene cluster was significantly low in individuals with high levels of fasting insulin ($P = 2.6e$-04) and serum triglyceride ($P = 3.0e$-05) levels (Fig. 6b). In addition, a significant negative correlation between the relative inositol pathway abundance with fasting insulin was also observed ($P = 1.1e$-03) (Supplementary Fig. 12). This observation is in line with a previous report on the association of *myo*-inositol pathway genes of *A. hadrus* SSC/2 located on a genomic structure variant (Supplementary Fig. 5) with lower host metabolic disease risk[23].

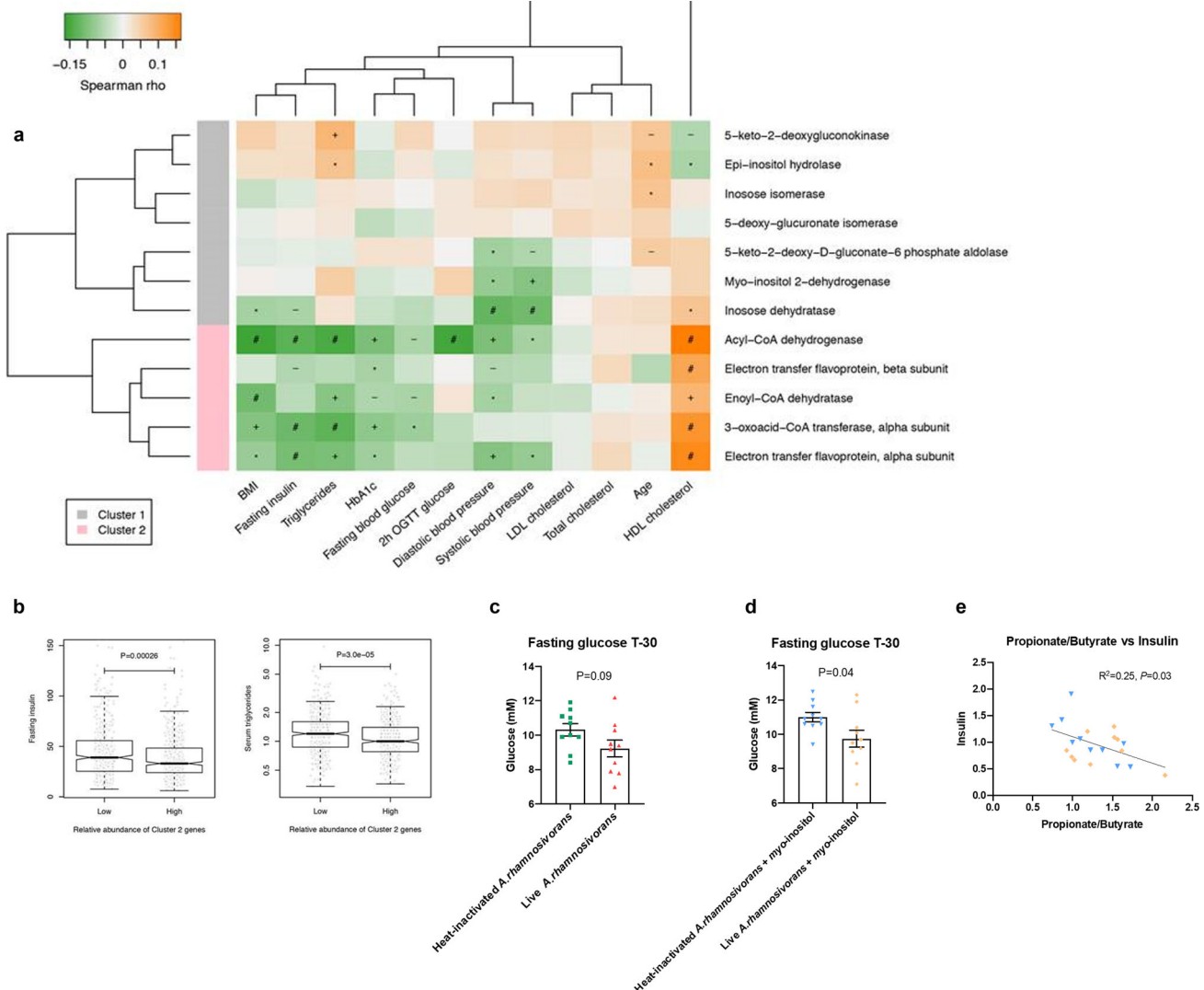

**Fig. 6 Propionate production gene cluster negatively associates with metabolic risk markers in prediabetes/diabetes cohort and reduced fasting glucose in mice fed with live *A. rhamnosivorans*.** Metagenomic analysis (**a**) shows that inositol utilisation gene cluster (cluster 1) did not form significant correlation with metabolic biomarkers while propionate production gene cluster (cluster 2) was significantly negatively associated with metabolic risk markers, especially fasting insulin and serum triglyceride (Spearman correlation); (**b**) shows significant differences in levels of fasting insulin ($P = 2.6\text{e-}04$) and serum triglyceride ($P = 3.0\text{e-}05$) in individuals with low versus high abundances of cluster 2 genes in a Swedish prediabetes cohort (Wilcox rank-sum test). $-P < 0.1$; $*P < 0.05$; $+P < 0.01$; $\#P < 0.001$. The horizontal line in each box represents the median, the top and bottom of the box the 25th and 75th percentiles, and the whiskers 1.5 times the interquartile range. Fasting glucose levels in western diet fed C57BL/6J mice after 6 weeks oral administration of heat-killed or live *A. rhamnosivorans* without (**c**) or with *myo*-inositol (**d**). $N = 10$ mice per group. Data are mean ± SEM; statistical analysis was done by unpaired two-tailed Student's *t* tests. Pearson correlation (**e**) between caecal propionate:butyrate ratio and fasting insulin in mice treated with heat-killed (blue circles) or live (orange circles) *A. rhamnosivorans* plus *myo*-inositol ($P = 0.03$).

Altogether these results indicate that propionate production from inositol by *Anaerostipes* species might provide potential health benefits.

To investigate the potential health benefit of *Anaerostipes* species we studied the impact of *A. rhamnosivorans* and inositol administration in mice fed a western diet (high in fat and sucrose)[59]. First, we tested the usefulness of this mouse model by determining the fate of live cells of *A. rhamnosivorans* when administered via oral gavage. The results show that *A. rhamnosivorans* persisted well in faecal samples of mice fed with a western diet (Supplementary Fig. 13). In this mouse model we subsequently administered live or heat-killed cells of *A. rhamnosivorans* in the presence or absence of *myo*-inositol for 7 weeks. Body weight gain did not differ between treatment

groups (Supplementary Fig. 14a–b) and neither did glucose tolerance determined after 6 weeks treatment (Supplementary Fig. 14c–f). We noted a trend ($P = 0.09$) towards lower fasting glucose after 6 weeks of treatment with live *A. rhamnosivorans* compared to heat-killed bacteria (Fig. 6c) and a significant ($P = 0.04$) lower fasting glucose when the mice were also treated with *myo*-inositol (Fig. 6d). In addition, the reduction of insulin levels was not significant between heat-killed and live treatment groups (Supplementary Fig. 14g–h) but we found a significant negative correlation ($P = 0.03$) between the ratio of caecal propionate: butyrate and fasting serum insulin levels for treatments of only co-administration of *myo*-inositol with either live or heat-killed *A. rhamnosivorans* (Fig. 6e). This suggests that microbial inositol conversion to propionate (rather than production of butyrate

from carbohydrates or lactate and acetate) may be needed to decrease insulin sensitivity. As we did not observe a strong effect of active treatment further investigations are required.

## Discussion

In this study, we describe a propionate synthesis pathway from *myo*-inositol by butyrogenic *Anaerostipes* species. This brings the number of propionate synthesis pathways up to four, including succinate, acrylate, propanediol and *myo*-inositol pathway. More than half of the tested *Anaerostipes* strains were able to grow on *myo*-inositol as the sole carbon and energy source by converting *myo*-inositol to propionate, acetate, formate, $CO_2$ and hydrogen. The *myo*-inositol pathway was further elucidated in *A. rhamnosivorans* DSM26241[T] by determining the metabolic route of the [13]C-labelled *myo*-inositol conversion by NMR, mass spectrometry combined with proteogenomic analysis and enzyme determinations. To our knowledge, it has not been reported elsewhere that intestinal bacteria are able to anaerobically convert *myo*-inositol into propionate. The predicted pathway includes *myo*-inositol catabolic genes which were to some extent similar to these in *Aerobacter aerogenes* and *Bacillus subtilis*[43–45]. However, one of the key steps of the propionate synthesis pathway that differentiated this from previously reported routes was the conversion of 3-oxopropionate to 3-oxopropionyl-CoA via a 3-oxoacid CoA transferase (Fig. 3). Interestingly, this latter enzyme based on the proteome data was predicted to be involved in propionate formation and this has been proven via enzyme activity assays (Fig. 4). Unlike the *myo*-inositol pathway in *A. rhamnosivorans* DSM26241[T], 3-oxopropionate was decarboxylated to produce acetyl-CoA that involved a CoA-dependent malonate-semialdehyde dehydrogenase in *Bacillus subtilis*[60]. The further conversion of 3-oxopropionyl-CoA to propionate is predicted to follow the reduction route and involve acyl-CoA dehydrogenase (AcaD encoded by AR1Y2_1117), enoyl-CoA hydratase (EcdH encoded by AR1Y2_1116), 3-oxoacid CoA transferase (OxcT encoded by AR1Y2_1115) acryloyl-CoA dehydrogenase and the α and β subunits of an electron transfer flavoprotein (AcaD-Etf encoded by AR1Y2_1050-1052) (Fig. 3a). *Myo*-inositol is naturally present in animal and plant cells in either free form or as a bound-component of phospholipids or inositol phosphate derivatives, such as phytate. The latter dietary compounds can release *myo*-inositol in the human colon via the activity of mainly microbial phytases in the intestine[61,62]. Our results on [13C_6] phytate enrichment using human stool samples show that intestinal bacteria are well capable of utilising phytate, to produce beneficial SCFAs to the human host (Fig. 5). Our data also indicate that phytate degradation by gut microbiota only leads to propionate and acetate formation in presence of *A. rhamnosivorans*. This is in line with the finding of a dietary intervention study where the abundance of *Anaerostipes* spp. was positively associated to an increase of faecal propionate formation in humans after a 3-month Mediterranean diet[41]. We discovered this *myo*-inositol pathway in *Anaerostipes* spp. that were previously only found to be colonic butyrate producers, illustrating the high metabolic flexibility of intestinal bacteria permitting quick adaptation to changes of nutrient supply in the gut.

Although the *myo*-inositol pathway was found in several *Anaerostipes* spp. (*A. rhamnosivorans*, *A. caccae*, *A. hadrus* PEL85), some *Anaerostipes* strains appeared not to have the *myo*-inositol pathway nor capability of fermenting *myo*-inositol (Fig. 1). This suggests that inositol fermentation might be obtained as the adaptation to substrate availability in the intestine. In fact, many copies of *myo*-inositol 2-dehydrogenase (IolG) detected in *A. rhamnosivorans* exemplified its adaptation to this

cyclic polyol. The detailed genomic analysis on *Anaerostipes* genomes showed that all *myo*-inositol degraders shared several gene clusters assembling the *myo*-inositol pathway (Supplementary Fig. 5). The gene arrangement of these clusters in the genomes was identical among *A. hadrus* strains and slightly different from that of *A. rhamnosivorans* and *A. caccae*, explaining the similar growth and metabolic activities on *myo*-inositol between *A. rhamnosivorans* and *A. caccae* (Fig. 1). Screening of over 10,000 NCBI genomes indicates that *Anaerostipes* is the key taxon in the gut that contains strains predicted to be capable of converting *myo*-inositol into propionate and acetate (Supplementary data 1). In addition, the metagenomic analysis revealed that the majority of analysed subjects carried the *myo*-inositol pathway genes, which is in line with the frequent detection of *Anaerostipes* in healthy humans. Moreover, we also found many subjects carried genes for phytases, indicating the prevalence of a food chain from dietary phytate involved *Anaerostipes* spp. in the gut. Importantly, the metagenomic analysis on the prediabetes/diabetes cohorts showed negative correlations of the propionate production gene cluster to metabolic disease biomarkers (Fig. 6a). This correlation was significant for fasting insulin and serum triglyceride in particular (Fig. 6b), indicating the potential benefit of inositol fermentation to propionate to host metabolism. We further investigated this by supplementation of *A. rhamnosivorans* with or without *myo*-inositol in mice. We found that live *A. rhamnosivorans* reduced fasting glucose as compared to heat-killed *A. rhamnosivorans* after 6 weeks and the reduction of glucose level became significant once *A. rhamnosivorans* was administered with *myo*-inositol (Fig. 6c, d), suggesting the observed reduction of fasting glucose level might be via active inositol fermentation of *A. rhamnosivorans*. Although no difference in neither insulin level nor glucose tolerance was observed, it has been shown that some intestinal bacteria need a much longer treatment with a high frequency in order to get the best effect while no effect was observed in a short treatment[63]. Therefore, our results provide the first indication of potential health benefits of *A. rhamnosivorans* and imply that further studies are needed to identifying the optimal treatment frequency, duration and dosage, as well as understanding the interactions between *A. rhamnosivorans* and host gut microbiota. Thus, it would be interesting to study the influence of environmental factors on the expression of the inositol pathway in *Anaerostipes* spp. in the gut.

A previous study reported that the presence of a microbial genomic structural variant of *A. hadrus* SSC/2 carrying *myo*-inositol pathway genes (Supplementary Fig. 5) was associated with reduced host metabolic disease risk[23]. In addition, a significant decrease of *Anaerostipes* abundance was also observed in type 2 diabetes patients compared to healthy subjects[22]. Our detailed genomic data analysis showed that the genomes of *A. hadrus* type strain DSM 3319[T] and *A. hadrus* DSM108065 do not predict the capacity to utilise *myo*-inositol, which was confirmed by our growth experiments. However, the *myo*-inositol pathway was detected in genomes of other *A. hadrus* strains that we found to produce propionate from *myo*-inositol (Fig. 1). This reportedly also included *A. hadrus* SSC/2 but as this strain is not publicly available we could not verify this experimentally. In addition, our results indicate that the proposed effect of *myo*-inositol fermentation is likely via the production of propionate rather than butyrate as previously suggested[23]. Supplementation of the human gut microbiome with *A. rhamnosivorans* may promote propionate production via inositol or phytate, hence lowering the risks of metabolic diseases. In fact, propionate has been proposed to have indirect effects in diabetes via lowering lipogenesis and serum cholesterol levels, as a result reducing the risk of diabetes development[64]. As oral *myo*-inositol supplementation has shown to improve insulin sensitivity in mice and human[36,37,65,66], our

data point out that the effect of administered *myo*-inositol may be via supporting the growth of *Anaerostipes* spp. in the intestine, resulting in propionate formation which is beneficial to human host. Although there are active *myo*-inositol transporters in the small intestine, *myo*-inositol uptake can be inhibited by relatively high mono-sugar load at this site[67], funneling leftover *myo*-inositol to the large intestine as substrate for bacterial fermentation. Besides, the combined supplementation of *chiro*- and *myo*-inositol has shown to have reduced the risk of metabolic disease compared to that of *myo*-inositol treatment alone[68]. It is conceivable that the greater effect of the combined treatment was due to the enhanced growth of *A. rhamnosivorans* thanks to its capability of metabolising both isomers. Altogether our strong metagenomic association data with the observed modest in vivo effects suggest the co-administration of both *myo*-inositol and *A. rhamnosivorans* as a potential therapeutic approach to improve host health but further investigations on optimal dosage, treatment duration and treatment frequency are needed.

## Methods

**Bacterial strains and growth media**. *Anaerostipes rhamnosivorans* 1y-2[T] was isolated by us previously and is available as DSM26241[T]. *Anaerostipes caccae* DSM14662[T], *Anaerostipes hadrus* DSM3319[T], *Anaerostipes hadrus* DSM108065 and *Anaerostipes butyraticus* DSM 22094[T] were obtained from German Collection of Microorganisms and Cell Cultures (DSMZ, Brunswick, Germany). *Anaerostipes hadrus* strain PEL85 was kindly provided by Dr Ulla Hynonen of the University of Helsinki, Finland.

All bacterial strains were routinely maintained in a modified YCFA medium supplemented with glucose. YCFA medium composed of (g/l): 10 yeast extract, 10 soy peptone, 4 sodium bicarbonate, 2,7 sodium acetate, 4,5 monopotassium phosphate, 0.9 dipotassium phosphate, 0.9 ammonium chloride, 0.9 sodium chloride, 0.09 magnesium sulfate, 0.0005 resazurin, 0.5 cysteine and 1 ml of vitamin mixture[69]. The vitamin mixture (per 100 ml) contains 1 mg biotin, 1 mg cobalamin, 3 mg 4-aminobenzoic acid, 5 mg folic acid and 15 mg pyridoxamine.

The growth experiments were performed in triplicate in 20 ml bicarbonate-buffered CP medium[50] supplemented with 20 mM *myo*-inositol or 10 mM *chiro*-inositol in a 50 ml anaerobic serum bottle filled with $N_2/CO_2$ (80:20, v/v) gas in the head phase. The inositol isomers were filter sterilised as stock solutions of 0.5 M. The condition in which the bacteria were added to 20 ml bicarbonate-buffered CP medium without substrate was used as control. The growth was monitored via metabolite formation by HPLC and optical density measurement by a spectrophotometer at a wavelength of 600 nm. To test the influence of acetate on inositol fermentation, the bacteria were grown in a CP medium with an addition of either 20 mM *myo*-inositol or 20 mM *myo*-inositol plus 20 mM acetate.

The coculture experiment between *Bifidobacterium longum* subsp. *infantis* DSM20088 and *A. rhamnosivorans* was performed in 20 ml YCFA medium supplemented with 20 mM sodium phytate (SIGMA). Sodium phytate was filter sterilised as stock solution of 0.5 M. The metabolite formation was analysed by HPLC. The monocultures of *B. longum* subsp. *infantis* and *A. rhamnosivorans* in the same medium were used as control. The experiment was performed in duplicate.

**Analytical methods**. *Myo*-inositol, *chiro*-inositol, glucose, rhamnose and short-chain fatty acids and alcohols were quantified on a Thermo Scientific Spectra HPLC system equipped with a Agilent Metacarb 67 H 300 × 6.5 mm column. The column was kept at 45 °C while running with 0.005 M $H_2SO_4$ as eluent under a flow of 1 ml/min. The detector was a refractive index detector. All analyses were performed in duplicate.

**Mass spectrometry**. Metabolic pathways and inositol intermediates formation were confirmed by liquid chromatography high-resolution tandem mass spectrometry experiments (LC-MS/MS): 10 mM inositol was added to two grown cultures of *A. rhamnosivorans* from sterile 1 M stock solution then bacteria were harvested at 30 min, and 1, 2, 3, 4 and 24 h. Metabolite production was stopped upon protein precipitation through the addition of a mixture of acetonitrile /methanol/ water (40/40/20, v/v/v); 10 ml of samples were filtered and sterilised through 0.2 μm syringe filters and the eluate collected for analysis. Data were acquired using a LTQ Orbitrap XL interfaced to an Ultimate 3000 RS (Thermo Fisher Scientific, Bremen, Germany). Tentative identification of *A. rhamnosivorans* inositol metabolites was achieved after two complementary trials: (I) reversed phase positive ion untargeted data dependent scanning mode for the identification of organic acid coenzyme A adducts; (II) hydrophilic interaction in targeted data dependent negative ions mode for polyalcohol intermediates.

The first setup included the use of a thermostated (35 °C) mixed-mode stationary reversed phase with positive charge surface (Luna Omega PS C18, 100

×2.1, 2.6 μm, Phenomenex, Torrance, CA) to improve the ionic interaction of organic acids- coenzyme A analytes. Mobile phases (flow 0.2 ml/min) consisted in 0.1% formic acid in water (solvent A) and 0.1% formic acid in methanol (solvent B). Samples (5 μl) were injected without any further dilution and analytes separated through the following gradient of solvent B (minutes/%B): (0/10), (2/10), (7.5/80), (9.5/80), while ESI interface parameters were the following: spray voltage 5.0 kV, capillary voltage 15.0 V, capillary temperature 300 °C, e sheath gas flow and auxiliary gas flow were 30 and 5 arbitrary units, respectively. Profile data type were acquired in full scan Fourier transformed high-resolution (FTMS) mode in the mass scanning range 120–900 *m/z*. For data dependent scanning mode, MS/MS normalised collision energy was set to 35, activation Q 0.25, activation time 25 ms, with a 1 *m/z* isolation window, while a reject mass list was generated by injecting blank samples.

The second setup was based on the use of a silica sulfobetaine zwitterionic modified HILIC column (100 ×2.1 mm, 1.7 μm, Thermo Fisher) at 35 °C. Mobile phases consisted in 0.1% formic acid in acetonitrile (solvent A) and 0.1% formic acid in water (solvent B). Undiluted samples were separated through the following gradient of solvent B (minutes/%B): (0/10), (2.5/10), (8.5/85), (10/85) at a flow rate of 0.2 ml/min and the injection volume was 5 μl. The current ion of potential candidates was scanned in the *m/z* range of 60–300 in negative mode and the ion source parameters were the following: spray voltage −4.5 kV, capillary voltage −25.0 V[70], sheath gas and auxiliary gas were 25 and 10 arbitrary units, respectively. Target analytes listed in Supplementary Table 4 were scanned in FTMS and MS/MS experiments by using CID fragmentation with a normalised collision energy set to 30; wherever target analytes were not detected, for data dependent scanning mode, MS/MS normalised collision energy was set to 35, activation Q 0.25, activation time 25 ms, with a 1 *m/z* isolation window, while a reject mass list was generated by injecting blank samples. Analyte responses were monitored by using Xcalibur 2.1 (Thermo Fisher Scientific, Bremen). While target analytes chemical composition, high-resolution spectra and tandem mass spectra were compared with reference compounds present in publicly available databases as Human Metabolome Database (https://hmdb.ca/) and KEEG (https://www.kegg.jp/) and the reliability of the spectral assignment to the metabolite was achieved according to Metabolomics Standards Initiative (MSI) level[70].

**Nuclear magnetic resonance**. *A. rhamnosivorans* strain 1y-2[T] was cultivated in a bicarbonate-buffered medium containing 10 mM of [13C]*myo*-inositol, 7 mM of [4-13C]*myo*-inositol or 7 mM of [4,5-13C₂]*myo*-inositol. The growth conditions were as described above. 13C-labelled *myo*-inositol and 13C-labelled phytate were synthesised and purified as previously reported[71]. Samples were taken during the growth phase at several timepoints and centrifuged at 10,000×g. Supernatants were amended with 0.5 ml D₂O (50 μl, 99.9 atom%, Sigma Aldrich) and subsequently transferred in NMR tubes (Campro Scientific). 13C-NMR spectra were recorded at a Bruker Avance 600 III HD spectrometer equipped with a PA BBO 600S3 BB-H-D-05-Z NMR probe, operating at 600 MHz (14T) at MAGNEtic resonance research FacilitY (MAGNEFY, Wageningen University and Research, the Netherlands). To achieve an optimal signal to noise ratio, a total of 256 scans were recorded for all samples, except for the samples containing 7 mM of [4-13C]*myo*-inositol that were measured with a total of 1024 scans due to low concentrations of labelled products. Supernatants collected from faecal enrichments on [13C₆]phytate in the presence and absence of *A. rhamnosivorans* were analysed using the same setting as described above. Chemical shifts are expressed in ppm. The peak assignment was performed by running the pure compounds of [13C₆]phytate, [13C]*myo*-inositol, [1-13C]propionate, [2,3-13C₂]propionate, [1-13C]acetate, [13C]CO₂, [2-13C]butyrate and these were also compared to these in Biological Magnetic Resonance Data bank (http://www.bmrb.wisc.edu/metabolomics/metabolomics_standards).

**Phytate degradation by gut bacteria**. Fresh stools were collected from two healthy donors of whom informed consents were obtained following Good Clinical Practice. We have complied with all relevant ethical regulations for work with human participants. The stools were freshly diluted in anaerobic PBS before being used as inoculum in a bicarbonate medium containing ~7 mM 13C-phytate as sole substrate either supplemented without or with *A. rhamnosivorans* (2% inoculum). Bicarbonate-buffered medium containing only [13C₆]phytate or [13C₆]phytate with *A. rhamnosivorans* were used as controls. The samples were taken after inoculation and 4 days incubation. The supernatants of these samples were used for 13C-NMR analysis to monitor substrate consumption as well as HPLC analysis for metabolite production while the pellets were harvested for gDNA extraction and followed by quantitative PCR for *A. rhamnosivorans* cell counts. As the 16S rRNA sequence of *A. rhamnosivorans* is highly similar to that of *A. caccae*[28], qPCR primers for *A. rhamnosivorans* were designed to have two nucleotides different from that of *A. caccae*[72], allowing the primers to amplify 16S rRNA sequence of *A. rhamnosivorans* but not that of *A.caccae* and the rest. The primers were validated and optimised using gDNA of *A. rhamnosivorans* and *A. caccae*. Primers were A.rham_F: 5′-CTGCACTCTAGCATTACAGT-3′ and A.rham_R: 5′-GCGTAGGTGGCATGAT AAGT-3′, which resulted in a 83 bp amplicon. The 16S rRNA gene of *A. rhamnosivorans* was used to optimise temperature and make standard curves. The qPCR programme was 95 °C for 5 min and 35 cycles consisting of 95 °C for 30 s, 60 °C for

10 s and 72 °C for 40 s; 95 °C for 1 min and 60 °C for 1 min. DNA copies were calculated based on standard curves.

**DNA extraction, sequencing and annotation.** The extraction of genomic DNA of *A. rhamnosivorans* DSM26241[T] was performed using the MasterPure Gram-positive DNA purification kit (Epicentre) according to manufacturer's instructions. The quality of the extracted DNA was measured using a NanoDrop 2000 spectrophotometer (Thermo Scientific, USA). The genome was sequenced using a PacBio RSII instrument, and the data processing and filtering was done with PacBio SMRT analysis pipeline v2.2 and the Hierarchical Genome Assembly Process (HGAP) protocol (http://www.pacb.com/devnet/). The genome annotation was obtained by running the assembly on RAST server[73]. The function of proteins was predicted and verified manually by BLASTing the amino acid sequences in Pfam[74], InterPro[75], Brenda, Uniprot databases. The annotated genome of *A. rhamnosivorans* strain DSM26241[T] was deposited on NCBI under accession number (GCA_005280655.1).

**Genomic and metagenomic analysis.** Genomes of *Anaerostipes caccae* DSM 14662[T] (GCA_000154305.1), *Anaerostipes hadrus* DSM3319[T] (GCA_000332875.2), *Anaerostipes hadrus* strain BPB5 (NZ_CP012098), *Anaerostipes hadrus* SSC/2 (GCA_001998765.1) and *Anaerostipes hadrus* strain PEL85 (JYFK01000000) were retrieved from NCBI database. The homologues of inositol pathway were identified by blasting amino acid sequences of these proteins from *A. rhamnosivorans* strain 1y-2[T] on the genomes of other strains. The heatmap was made with seaborn python. Pairwise genome comparison was done on JSpeciesWS server[76].

Pre-quality filtered metagenomic reads obtained from stool of 65 Human Microbiome Project subjects were downloaded from RAST using its API. These subjects have been reported in previous study[50]. After converting reads to fasta format read alignment against the pathway proteins was performed with Diamond (v0.9.22) using a cut-off *e*-value of -e 0.000000001. Heatmap was made with Python3.8.2, the Seaborn library and Inkscape. Pathway abundance was calculated as #reads-pathway/((#reads-total × length-pathway-bp)/(4 × 10^6 bp))[51]. Kraken2 (2.0.9-beta) was used for taxonomic read classification in the metagenomic samples[77]. Kraken2 classification was run with – minimum-hit-groups of 2 and – report. The Kraken2 database used was built in July 2020 comprising of the complete genomes in RefSeq for the bacterial, archaeal and viral domains, along with the human genome and a collection of known vectors (UniVec_Core). Abundance estimation of the genus *Anaerostipes* (taxonomy ID 207244) was determined by Bracken (v2.5)[78]. The Bracken kmer distribution was built using the previous mentioned Kraken2 database with a kmer size of 35 and ideal read length of 350. Ideal read length was based on the median read length in all the metagenomic samples. The script est_abundance.py from Bracken was run with the previous mentioned kmer distribution, minimum read threshold of 1 and the Kraken2 report file.

**Proteomics.** *A. rhamnosivorans* was grown in two conditions including YCFA supplemented with 40 mM rhamnose or 40 mM *myo*-inositol. The experiments were performed in four replicates. The bacteria were harvested during exponential phase. The protein abundances in cultures growing with two different substrates were examined with liquid chromatography–mass spectrometry/mass spectrometry (LC-MS/MS). *A. rhamnosivorans* was grown in 4 replicates in 100 ml bicarbonate-buffered medium containing with 40 mM *myo*-inositol or 40 mM rhamnose as carbon and energy sources. Cells were collected in the exponential phase by centrifugation at 4700 × *g* at 4 °C for 20 min. Cell pellets were washed twice and suspended in 0.5 ml of 100 mM Tris-HCl (pH 8) and subsequently disrupted by 5-min sonication for 1 s and 2 s pause in between while cooling on ice in a Branson sonicator with amplitude of 50%. These suspensions were then centrifuged at 10,000 × *g* at 4 °C for 10 min to remove cell debris. The supernatants were reduced with 15 mM dithiothreitol and proteins were unfolded with 6 M urea and alkylated in 20 mM acrylamide before being washed twice with 50 mM ammonium bicarbonate by centrifugation of the buffer and reactants through a Nanosep 3K Omega filter (Pall) at 13500 × *g* for 30 min. Proteins were digested by addition of 0.5 μg sequencing grade trypsin in 100 μl of 50 mM ammonium bicarbonate and overnight incubation at room temperature. Peptides were acidified with 10% trifluoroacetic acid to pH 3 and centrifuged through the membrane. After the digestion, the samples were kept at −20 °C upon the analysis. The samples were analysed by injecting 18 μl sample over an analytical column 0.10 × 250 mm ReproSil-Pur 120 C18-AQ 1.9 μm beads (prepared in-house), with an acetonitrile gradient at a flow of 0.5 μl/min on a Thermo EASY nanoLC1000 (Thermo). The gradient consisted of an increase from 9 to 34% acetonitrile in water with 0.1% formic acid in 50 min followed by a fast increase in the percentage acetonitrile to 80% (with 20% water and 0.1% formic acid in both the acetonitrile and the water) in 3 min as a column-cleaning step. An electrospray potential of 3.5 kV was applied directly to the eluent via a stainless steel needle fitted into the waste line of the micro cross. Full scan positive mode FTMS spectra were measured between *m/z* 380 and 1400 on a Q-Exactive HFX (Thermo electron, San Jose, CA, USA) at resolution of 60,000. HCD MS/MS fragmentation scans of the 25 most abundant 2–5+ charged peaks in the MS scan were recorded in data dependent mode.

LC-MS/MS runs with all MS/MS spectra obtained were analysed with MaxQuant 1.6.3.4[79] using the "Specific Trypsin/P" Digestion mode with maximally 2 missed cleavages and further default settings for the Andromeda search engine except that Propionamide (C) was set as a fixed modification and variable modifications were set for de-amidation of N and Q, Protein N-terminal Acetylation and M oxidation[80]. *A. rhamnosivorans* database from NCBI (GCA_005280655.1) was used together with a contaminants database that contains sequences of common contaminants. The "label-free quantification" and the "match between runs" options were enabled. We used de-amidated peptides for protein quantification and all other quantification settings were kept default.

Filtering and further bioinformatic analysis of the MaxQuant/Andromeda workflow output and the analysis of the abundances of the identified proteins were performed with the Perseus 1.6.2.1 module (available at the MaxQuant suite). Peptides and proteins with a false discovery rate (FDR) of <1% and proteins with at least two identified peptides of which at least one should be unique and at least one should be unmodified were accepted. Reversed hits were deleted from the MaxQuant result table as well as all results showing a normalised label-free quantitation intensity (LFQ) value of 0 for all inositol and rhamnose injections. The normal logarithm was taken from protein LFQ MS1 intensities as obtained from MaxQuant. Zero "Log LFQ" values were replaced by a value of 5.9 (a value slightly lower than the lowest measured value) to make sensible ratio calculations possible. Relative protein quantitation of inositol-grown to rhamnose-grown cells was done with Perseus by applying a two sample Ttest using the "LFQ intensity" columns obtained with FDR set to 0.05 and S0 set to 1. nLC-MS/MS system quality was checked with PTXQC[81] using the MaxQuant result files.

**Phylogenetic tree construction.** A CoA transferase phylogenetic tree was constructed from butyryl-CoA:acetate CoA transferases (but), butyryl-CoA:4-hydroxybutyrate CoA transferases (4Hbt), butyryl-CoA:acetoacetate CoA transferase (Ato_B), butyryl-CoA:acetoacetate CoA transferase (Ato_A) from known intestinal butyrate-producing species that were retrieved from the NCBI database (Fig. 4). 3-oxoacid/propionate CoA transferases from *Coprococcus catus* (WP117539849); *Megasphaera elsdenii* (ALG42129); propionate CoA transferase from *Ralstonia eutropha* H16 (A2718)[53] and *Clostridium propionicum*_Propionate_CoA transferase (AJ276553)[52] were retrieved from NCBI and included in the tree. Amino-acid sequences of butyryl-CoA:acetate CoA transferase and 3-oxoacid CoA transferase from strain *A. rhamnosivorans* 1y-2[T] were aligned with retrieved sequences using the CLUSTAL_X programme. A phylogenetic tree was constructed using the neighbour-joining algorithm by the MEGA 7 with 1000 bootstraps to obtain confidence levels for the branches.

**Preparation of cell free extracts.** *A. rhamnosivorans* was grown in 10 ml anaerobic bicarbonate-buffered CP medium containing 20 mM rhamnose or 20 mM *myo*-inositol as carbon and energy sources. Cells of *Intestinimonas butyriciproducens* AF211[50] grown in 20 ml anaerobic bicarbonate-buffered CP medium containing 20 mM lysine were used as a negative control of the enzyme assay. Cells were collected at early and late exponential phase by centrifugation at 10,000×*g* at room temperature for 10 min. Cell pellets were washed twice in an anaerobic buffer containing 100 mM Tris-HCl (pH 8) and suspended in 0.5 ml of the same buffer. Cells were disrupted by repeated (5 times) sonication for 30 s and 30 s pause in between while cooling on ice in a Branson sonicator with amplitude of 50%. Finally, the suspension was centrifuged for 10 min at 10,000×*g*. The cell-free extracts were transferred to 5 ml serum bottles sealed with rubber stopper in the anaerobic chamber for subsequent enzyme assays. All steps were performed in an anaerobic chamber with a N2/H2 (96:4, v/v) atmosphere, circulated over a palladium catalyst to eliminate traces of oxygen. To study the activity of 3-oxoacid CoA transferase toward propionyl-CoA during exponential phase, *A. rhamnosivorans* was grown in *myo*-inositol and cells were harvested every hour for obtaining cell-free extracts. Subsequent processing steps were followed as described above. This experiment was performed in duplicate.

**Detection of CoA transferase activity.** CoA transferase activity in crude enzyme extracts was determined using a spectrophotometric assay[55]. In the assay, acetyl-CoA formed from CoA derivatives and acetate was condensed with oxaloacetate, thus releasing CoA which reacted with 5,5′-dithiobis(2-nitrobenzoate) to make a yellow thiophenolate anion. The reaction mixture (1 ml) contained 100 mM Tris-Cl (pH 8), 200 mM acetate, 1 mM oxaloacetate, 0.15 U citrate synthase, 1 mM 5,5′-dithiobis(2-nitrobenzoate) and either 20 μM butyryl-CoA or 20 μM propionyl-CoA. The reaction was performed in an anaerobic cuvette with N2 in the head space, kept at 25 °C. The formation of acetyl-CoA from butyryl-CoA or propionyl-CoA and acetate was based on the measurement of the yellow anion at 412 nm ($\varepsilon = 13.6$ mM$^{-1}$ cm$^{-1}$), which is produced via the coupling reaction. The activities were expressed as U mg$^{-1}$ of total proteins ($U = \mu$mol min$^{-1}$). The comparative enzyme assay between inositol and rhamnose conditions was replicated in a biological replicate, while the activity assay using extracts of cells grown in *myo*-inositol during exponential phase were also replicated in a biological duplicate with at least three technical replicates. Total protein concentration in the crude enzyme extracts was measured using Bradford assay according to the manufacturer's instructions (Pierce).

**Metagenomic analysis in prediabetic/diabetic cohorts and pathway screening**. BlastKOALA tool[82] was used to obtain the KEGG ortholog (KO) annotation for the two clusters of genes involved in metabolism of myo-inositol to propionate. The MEDUSA pipeline was then used for mapping the reads to those genes assigned to the aforementioned KOs in the IGT gene catalogue[58] to obtain the KO relative abundance table. The KO relative abundances were then associated with measured clinical biomarkers based on Spearman correlation analysis. BLASTP searches against the NCBI reference genomes[83] using query sequences identified from *A. rhamnosivorans* were performed for screening bacteria bearing the whole myo-inositol metabolism pathway. The following BLASTP search settings with sequence identity ≥50, query sequence coverage ≥70%, and E-value ≤ 1e-5 were used except for '3-oxoacid CoA transferase'; notice that only alignments with identity ≥70 were kept for this protein since several other highly homologues proteins were known. Because inositol utilisation gene cluster (cluster 1) genes in general have much broader bacterial distribution than cluster 2 genes, only species containing at least the entire propionate production gene cluster (cluster 2) were shown in the final list.

**Animal studies**. All animal experiments were done in male C57Bl/6J mice (JAX™ Mice Strain, Charles River Laboratories, Germany). Mice were housed in individually ventilated cages (Green line Sealsafe plus, Tecniplast, Buguggiate, Italy) with a maximum of five mice per cage. Mice were kept under constant temperature (19–21 °C); humidity of 40–70% and a 12-h light/dark cycle with free access to food and water. All animal experiments were conducted in accordance with the principles of the 'Guide to the Care and Use of Experimental Animals' were approved by the Ethics Committee on Animal Care and Use in Gothenburg, Sweden. We have complied with all relevant ethical regulations for animal testing and research.

To ensure colonisation of *A. rhamnosivorans* independent of diet, mice were fed either autoclaved chow diet (5021, LabDiet, St. Louis, MO) or irradiated western diet (WD; 41% kcal fat, 41% kcal carbohydrates TD.09683, Envigo). At the same time the mice were fed WD ($n = 5$ mice) or kept on chow diet ($n = 3$ mice), the mice were given 200 μl gavage of comprising $10^9$ CFU of Live *A. rhamnosivorans* in 10% glycerol stock three times during 1 week. Colonisation was examined in caecal samples collected 3 days after the last gavage.

To test if oral administration of *A. rhamnosivorans* could affect glucose metabolism and body weight in WD fed mice, mice were gavaged three times per week during seven weeks with either 0.2 ml of $10^9$ CFU of Live *A. rhamnosivorans* in 10% glycerol stock ($n = 10$ mice) or 0.2 ml of heat-killed *A. rhamnosivorans* ($n = 10$ mice). To investigate if combination of *A. rhamnosivorans* and myo-inositol could have a synergistic effect, myo-inositol (0.1 mg/g body weight) was added to the bacterial suspensions of live ($n = 10$ mice) and heat-killed ($n = 10$ mice) *A. rhamnosivorans* at the same dose regimen as described above. Mice were weighed every week. An oral glucose-tolerance test (2 g/kg body weight) was done after 6 weeks of treatment. Mice were fasted for 5 h and blood glucose was measured from tail blood samples at $t = 0$, 15, 30, 60, 90 and 120-min after glucose gavage. Tail blood was also collected for insulin analysis (at $t = -30$ min, 15 min and 30 min) and analysed with the Ultra-Sensitive Mouse Insulin ELISA kit (Crystal Chem, Downers Grove, IL, USA) according to the manufacturer's protocol. After 7 weeks of treatment, mice were fasted for 5 h and blood and tissues were harvested. Caecum short-chain fatty acids were measured using gas chromatography coupled to mass spectrometry detection (GC-MS). Briefly, ~20–50 mg of caecum content were mixed with internal standards (16 mM acetate, 3.2 mM propionate and 3.7 mM butyrate), added to glass vials and freeze dried. All samples were then acidified with 50 μl of 37% HCl, and SCFAs were extracted with two rounds of diethyl ether extraction (2 ml diethyl ether, rotation shake for 15 min, and centrifugation for 5 min at 2,000×g). The organic supernatant was collected; 50 μl of the derivatization agent N-tert-butyldimethylsilyl-N-methyltrifluoroacetamide (Sigma-Aldrich) was added and samples were incubated at room temperature overnight. SCFAs were quantified with a gas chromatograph (Agilent Technologies 7890A) coupled to a mass spectrometer (Agilent Technologies 5975C). Short-chain fatty acid standards were attained from Sigma-Aldrich (Stockholm, Sweden).

**Reporting summary**. Further information on research design is available in the Nature Research Reporting Summary linked to this article.

## Data availability

The whole-genome sequence of *A. rhamnosivorans* DSM26241$^T$ has been deposited in NCBI under the BioProject accession number PRJNA540423 and SRA accession number CP040058. The raw mass spectrometry proteomics data have been deposited to the ProteomeXchange Consortium via the PRIDE repository with the dataset identifier PXD021084. The entire proteome is provided in the Supplementary information/source data file. Metagenomes of Swedish Impaired Glucose Tolerance (IGT) cohort were used for pathway analyses (https://doi.org/10.1016/j.cmet.2020.06.011). The clinical data are available under restricted access for ethical and legal considerations, access can be obtained following the standard application to the data committee. Other associated data generated in this study are provided in the Supplementary Information/Source Data file. The raw metagenomes data used in this study are available in the China NGDC Genome Sequence Archive under accession code HRA000020. Used databases are biological Magnetic Resonance Data bank (http://www.bmrb.wisc.edu/metabolomics/

metabolomics_standards); Pfam (http://pfam.xfam.org/), InterPro (https://www.ebi.ac.uk/interpro/), Brenda (https://www.brenda-enzymes.org/), Uniprot (https://www.uniprot.org/); NCBI genome (https://www.ncbi.nlm.nih.gov/genome/). Human Metabolome Database (https://hmdb.ca/), KEGG (https://www.genome.jp/kegg/compound/). Source data are provided with this paper.

## Code availability

A custom code for metagenomic analyses has been deposited in github (https://gitlab.com/bartn/inositol) and a doi (https://doi.org/10.5281/zenodo.5062693)

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

## Acknowledgements

This work was partially supported by the SIAM Gravitation Grant 024.002.002 and Spinoza Award of the Netherlands Organization for Scientific Research to WMdV as well as Caelus Pharmaceuticals B.V. Authors acknowledge Jarmo Ritari & Lars Paulin for their assistance for genome sequencing, Daan Vliet for his assistance in genome submission and Tatiana Nikolaeva for technical support for NMR analysis; Andrea for advice on LC-MS analysis; Anna Hallén for the assistance in gavage treatments in the mice.

## Author contributions

W.M.dV. and T.P.N.B. designed the research. T.P.N.B. performed the research. R.P. and D.F. synthesised and purified $^{13}$C-myo-inositols and $^{13}$C-phytate. L.H. performed the animal study. H.W. performed metagenomic/genomic analysis on (pre)diabetic cohorts. A.D.T. ran and analysed mass spectrometry for intermediate identification. B.N. contributed metagenome analyses, S.B. performed LC-MS/MS for proteomics. T.P.N.B., L.H., H.W., A.D.T., B.N., S.B., F.B. and W.M.dV. analysed the data, T.P.N.B. and W.M. dV. wrote the paper. All authors commented and approved for the manuscript.

## Competing interests

The Wageningen University has applied for a patent relating to the use of bacteria capable of converting inositol into propionate on which T.P.N.B. and W.M.dV. are inventors (Wageningen; WO2021028585). Other authors declare no competing interests.
