## [Peer Review File · Nature Communications]

Reviewers' Comments:

Reviewer #1:

Remarks to the Author:

Manuscript NCOMMS-20-31791 "A novel pathway for inositol conversion into propionate by *Anaerostipes* supports its potential health impact".

General comments.

This manuscript describes a very interesting topic. However, I find this study to a large degree lacking comprehensive biochemical evidence for the myo-inositol cascade as well as biological examples on how myo-inositol affects the gut microbiota. Furthermore, this study lacks supporting/comprehensive evidence for several claims which in many ways remain speculative, notably there are only 4 figures in the main text and NMR is the primary analytical method conducted to provide biochemical evidence. All growth experiments lack sufficient data depth with merely duplicate samples.

The introduction/background lacks balance, with referencing state of the art/existing literature being incomplete/superficial.

Specific comments.

L43. Is it well documented and consensus that inositol stereoisomers are promoting health? I'm not convinced, and authors does not provide support for such a claim. Furthermore, L44 implies that this conversion of phytate to propionate is ONLY performed by *Anaerostipes* spp. It is likely that other phyla could have other protein folds performing similar types of catalysis.

L385. This is a speculative. *Anaerostipes* could be stimulated by a myriad of other compounds (rather than myo-inositol) found in a Mediterranean diet. This is one example of a number of speculative statements throughout this manuscript.

Another example of speculation that to a little degree is supported by data occurs at line 417-425. The authors are discussing propionate increase when persons consumed whole grain, and implies that this is due to myo-inositol in the whole-grain, which obviously contain an arsenal of other components that could contribute to propionate production

There is a lack of biochemical evidence. E.g. I find it intriguing that there are no attempts of cloning the enzymes central in this pathway or produce knock outs of enzymes that are found to be central in this myo-inositol pathway, in order to demonstrate that the enzymes in question are essential for the myo-inositol conversion. Furthermore, I would suggest to set up a chromatographical method for the spent media and observe fermentations over time course. This would complement the NMR data, provide reasoning for the selected time points in the NMR experiments (which provide more in-depth analysis), and also be used for relevant control experiments.

Line 386-389. Authors begin sentence with "Remarkably" quick adaptation to changes in nutrient supply is nothing new, and has been shown by groups like Martens et al. on time adaptation of gut microbiota to various dietary fibers.

Fig. 1

Authors does not comment on the fact that acetate is produced to a similar extent as the propionate, why?

Fig. 2. Very nice and informative figure. However, what kind of control experiments are provided? Furthermore, there is a lack of complementary analyses to support the NMR evidence. E.g. evidence supported by chromatography or similar, on both C13 enriched and C12 samples would strengthen this study (and is essential) considerably. It would also allow methodology that are easier accessible than (high)-field-NMR, and make the experiments easier to validate.

Fig. 4.

a) I would have liked a phylogenetic tree displaying *Anaerostipes* spp in a wider context, to provide readers a broader overview. What about propionate producer from other Phyla??

b) Each measurement is conducted in triplicates, I understand this as technical replicates. What about the biological replicates?

c) For B-D) 2 timepoints is (very) meager. I would suggest to run spent media from cultures, to observe the utilization of myo-inositol during time-course, this would provide a higher time

resolution.

Reviewer #2:

Remarks to the Author:

This manuscript reports a novel pathway for inositol conversion into propionate by several strains of *Anaerostipes* spp., highlighting the potential beneficial effects of inositol stereoisomers and phytate on human metabolic health. The novel pathway was characterized by ¹³C-NMR and proteogenomics.

The paper is well written, the results are relevant and technically well-conducted. The main drawback is the numerous assumptions or potential scenarios ("likely", "suggest that might", "might be a potential", "might be related") that are not supported with direct experimental data.

Of note, in the elucidation of the myo-inositol fermentation pathway many relevant intermediates were not experimentally detected. The authors only focused on precursor and final products, elucidating (indirectly) the numerous intermediate reactions and metabolites (depicted in Fig 3A) from the proteogenomic analysis. This could be due to poor sensitivity of NMR or other biochemical reasons. There are more sensitive techniques, mainly mass spectrometry and isotopologue analysis, that could nicely complement C-NMR and isotopomer characterization. In my view, it would be important to detect and characterize the isotopic enrichment of, at least, 5-dehydro-2-deoxy-D-gluconate 6-phosphate, 3-oxopropionate and 3-oxopropionyl-CoA.

The same holds true for the physiological relevance in human gut:

It is stated that there is cross-feeding between *B. longum* subsp. *infantis* and *A. rhamnosivorans* on phytate via myo-inositol interspecies-transfer, leading to an increased propionate production, however this is not supported by direct quantitative stable isotope labeling data using ¹³C-phytate.

Similarly, it is stated that "human often consume phytate-containing plants such as brans and cereals, and hence myo-inositol is likely released by microbial activities in the large intestine and serves as substrate for intestinal bacteria", however no in vivo evidence is provided, either by administering ¹³C-inositol or ¹³C-phytate to mice (which also contains *Anaerostipes* spp in the gut microbiota) and detecting labeled propionate in feces and serum.

Overall, I'm in favor of publication in Nature Communications after the preliminary nature of some claims are supported with additional experimental data.

Reviewer #3:

Remarks to the Author:

In this study the authors examine a few strains of *Anaerostipes* in the presence of myo-inositol, and determine that a some species from this genus can produce propionate (Figure 1). They then go into figures/discussion that help build the hypothesis that these strains may uniquely express pathways that enable conversion of myo-inositol to propionate. The authors discuss how this could be biologically important given the effects shown by others for propionate/SCFAs. They further suggest that breakdown of phytate by distinct commensal bacteria may provide an additional inositol source. The ideas presented in the manuscript are interesting, and are supported, to some degree, with data. However, the study falls short of validating the ideas proposed with experimentation that directly tests the suggested hypotheses. Loss/gain of function studies and colonization experiments are needed to support the authors' suggested mechanisms and in vivo significance.

Some specific concerns:

1. The strains on the left of Figure 1 exhibit increased formate and propionate, in combination with decreased inositol. Thus it is clear why the author interpret that the bacteria metabolize inositol to propionate. However, these key initial studies lack necessary controls- including how

propionate levels compare in this time course of bacterial growth in the same media when no inositol is present. It's possible that the metabolite changes are independent and this can not be known without this control. In addition, do propionate levels remain elevated once inositol is depleted? If so, for how long. The time course should be extended.

2. The authors discuss the microbiome and the likely relevance of the pathway in vivo. However, the biologic relevance in vivo is not examined anywhere in the paper. Do these strains survive in the intestine? If so, mice should be colonized with a propionate producer (*A. rham*), and well as a non-producer (*A. hadreus*) and metabolite levels compared to germ-free mice (with and without inositol).

3. Figure 3A depicts the inositol conversion pathway and *A. rhamnosivorans* produces proteins that would be predicted to enable this pathway. However, its unclear how values compared to other strains of bacteria that are not predicted to utilize this pathway. Figure 3C mostly demonstrates similarities across the strains. However, the assumption is that the 2nd row (Myo-inositol 2 dehydrogease) is the key difference between the inositol to propionate producer (*A. rham*), versus a non-producer (*A. hadreus*). This is a very big leap. Potential genes driving this difference were discussed in the text, but they should be manipulated in order to test the hypothesis that these proteins are the molecular mechanisms underlying the phenotype.

4. Despite extensive discussion of phytate, the only figure with phytate is Supplemental Figure 6 which is quite weak data. Further, it is unclear why *A. rhamnosivorans* alone has high acetate and butyrate- this is not consistent with Fig. 1. In addition, the study lacks the key control comparing to media without phytate. Thus, it lacks evidence that any differences are due to phytate metabolism. This direction of the study seems tangential and unnecessary. Co-colonization in vivo would be necessary to support proposed conclusions.

5. There are no statistics shown in any figure, or described.

Minor.

Some inaccuracies between the figure referenced and order in the text were noted.

REVIEWER COMMENTS

Reviewer #1 (Remarks to the Author):

Manuscript NCOMMS-20-31791 "A novel pathway for inositol conversion into propionate by *Anaerostipes* supports its potential health impact".

General comments.

This manuscript describes a very interesting topic. However, I find this study to a large degree lacking comprehensive biochemical evidence for the myo-inositol cascade as well as biological examples on how myo-inositol affects the gut microbiota. Furthermore, this study lacks supporting/comprehensive evidence for several claims which in many ways remain speculative, notably there are only 4 Fig.s in the main text and NMR is the primary analytical method conducted to provide biochemical evidence. All growth experiments lack sufficient data depth with merely duplicate samples. The introduction/background lacks balance, with referencing state of the art/existing literature being incomplete/superficial.

Thank you for your suggestions and comments. We have performed additional experiments to detect intermediates of the inositol pathway by sensitive LC-MS/MS, repeated all biochemical and growth experiments, and included new experiments with [¹³C₆]phytate, results of advanced metagenomic analysis, and beneficial effects of feeding *A. rhamnosivorans* and myo-inositol in insulin-resistant mice fed a western diet (New Fig. 5 and 6). We also rewrote the Introduction with references to the state of the art. This is detailed below.

The myo-inositol cascade till to dihydroxyacetone has been described previously and the enzymes involved in this part of the pathway have been biochemically and genetically characterized in *Lactobacillus casei* (Yebrá *et al.*, 2007) and *Bacillus subtilis*, where these are annotated as products of the *iolABCDEFGHIJ* operon (Yoshida *et al.*, 2008). This is now included in the Introduction (Lines 120-123). Hence, the identified genes in this part of the pathway in *Anaerostipes rhamnosivorans* are homologs of those from *B. subtilis*. The encoded proteins are highly abundant and over-produced under inositol growth condition in *A. rhamnosivorans* as compared to rhamnose condition (see Fig. 3A-B). Hence, we believe that this is sufficient support for the involvement of these enzymes in the pathway that is supported by the detection of the intermediates scyllo-inosose, 3,5/4-trihydroxycyclohexa-1,2-dione, 5-dehydro-2-deoxy-D-gluconate and 5-dehydro-2-deoxy-D-gluconate 6-phosphate (see below) by sensitive LS-MS/MS.

The novel part of pathway in *A. rhamnosivorans* includes the fate of 5-dehydro-2-deoxy-D-gluconate 6-phosphate and here we have provided biochemical evidence for the (induced) enzyme involved in the conversion of 3-oxopropionate to 3-oxopropionyl-CoA (see Fig. 4B-F). These experiments were repeated and confirm our previous conclusions. Likely due to low half-life times, we were not able to detect further intermediates with the sensitive LC-MS/MS technique applied but have ample evidence for the formation of the labeled end-products propionate and acetate by the experiments with ¹³C labeled inositols.

To improve the data depth for the growth experiments, we performed these again in triplicate with more time points and visualized the results in the revised Fig. 1 that included growth curves of all tested *Anaerostipes* strains.

We also added two more Figures (Fig. 5 and 6) providing additional evidence for the physiological relevance of phytate/inositol fermentation in the gut and the potential impact of *Anaerostipes* on host health in the main text. Fig. 5 shows the results from [¹³C₆]phytate enrichments using fecal samples, confirming that propionate production from phytate degradation occurs only when *Anaerostipes* was present.

Fig. 6 represents the results from the metagenomic analysis of inositol pathway genes in prediabetes/diabetes cohorts of which negative correlations between a propionate production gene cluster (cluster 2) and metabolic biomarkers were clearly observed in prediabetic/diabetic cohorts, especially fasting insulin and serum triglyceride (Fig. 6B). In response to the remarks of reviewers 2 and 3 (see below) we performed an experiment with insulin-resistant mice on a western diet (Koh *et al.*, 2020). We observed that these mice fed with live *A. rhamnosivorans* had lower fasting glucose levels as compared to mice fed with heat-killed *A. rhamnosivorans*. The reduction of fasting glucose level became significant once *A. rhamnosivorans* was supplemented together with *myo*-inositol. We also found that a ratio of cecal propionate/butyrate was negatively associated with insulin levels in mice fed with bacteria and *myo*-inositol, implying the potential link of active *myo*-inositol fermentation to propionate with lowering insulin level in mice. Our results indicate the potential benefit of *A. rhamnosivorans* on reducing glucose level in mice, possibly via active inositol fermentation leading to the production of beneficial propionate.

To provide better clarity of existing literature in the introduction, we have added more references and modified the text in the revised version. All changes are highlighted.

Specific comments.

L43. Is it well documented and consensus that inositol stereoisomers are promoting health? I'm not convinced, and authors do not provide support for such a claim. Furthermore, L44 implies that this conversion of phytate to propionate is ONLY performed by *Anaerostipes* spp. It is likely that other phyla could have other protein folds performing similar types of catalysis.

Thank you for these questions. The positive effects of inositol stereoisomers on host health, especially improving insulin resistance, have been reported in a numerous studies and this has been shown in mice, monkey and human (Ortmeyer *et al.*, 1995, Nordio & Proietti, 2012, D'Anna *et al.*, 2013, Croze *et al.*, 2015, Pintaudi *et al.*, 2016). We have cited these all in the Introduction. We specifically want to point to studies where *myo*-inositol supplementation improves insulin resistance in patients with gestational diabetes (Corrado *et al.*, 2011). However, we agree with the reviewer that not all cited studies are equally strong and hence we have tuned these down in the Introduction by using the term 'suggested' (please see line 101). Moreover, in the abstract we have inserted the term 'potential' as to provide the nuance that is alluded to by this reviewer. Based on the mouse data provided in our study, we may speculate that the health benefits of inositol-stereoisomers may depend on the level of inositol-degrading *Anaerostipes* spp. in the gut of subjects recruited in these intervention studies.

The reviewer asks appropriately for the evidence for our suggestion that the conversion of phytate to propionate is ONLY performed by *Anaerostipes* spp. Of course, we cannot completely rule out other taxa but we now screened over 10 000 available genomes in the NCBI database for the presence of the inositol to propionate fermentation pathway and found this complete pathway only in *Anaerostipes* and very related (gut) species (Supplementary data 1). In the experimental data we also showed that *Anaerostipes* spp. used this metabolic feature to cross-feed with Bifidobacteria on phytate, which resulted in an increase of

beneficial propionate. On top of this, we performed additional experiments where stools were inoculated in a medium supplemented with [¹³C₆]phytate with and without *A. rhamnosivorans*. The data show that [¹³C]propionate was only formed in presence of *A. rhamnosivorans*, confirming our hypothesis on propionate production derived from *A. rhamnosivorans* during phytate degradation. These data have been included in Fig. 5 in the revised version of the manuscript.

We have rewritten the abstract and mentioned this in the text as “These results suggest that *Anaerostipes* is a key taxonomic group, members of which are predicted to contain the entire metabolic pathway to convert inositol to propionate”. Please see lines 404-405.

L385. This is a speculative. *Anaerostipes* could be stimulated by a myriad of other compounds (rather than myo-inositol) found in a Mediterranean diet. This is one example of a number of speculative statements throughout this manuscript.

We agree with the reviewer that this is speculative among others as inositol intake was not determined. However, since we observed propionate formation from phytate degradation by human microbiota in presence of *Anaerostipes* in our study, we would like to mention this study as an example of dietary interventions that may stimulate *Anaerostipes* growth/metabolism by providing phytate derived substrates. Therefore, the text has been tuned down and modified in the revised version. Please see lines 471-474 that read like: ‘This is in line with the finding of a dietary intervention study where the abundance of *Anaerostipes* spp. was positively associated to an increase of faecal propionate formation in humans after a 3-month Mediterranean diet⁴⁰’.

Another example of speculation that to a little degree is supported by data occurs at line 417-425. The authors are discussing propionate increase when persons consumed whole grain, and implies that this is due to myo-inositol in the whole-grain, which obviously contain an arsenal of other components that could contribute to propionate production

We agree with the reviewer that this study is speculation. Therefore we removed this in the revised version of the manuscript. Please see line 520.

There is a lack of biochemical evidence. E.g. I find it intriguing that there are no attempts of cloning the enzymes central in this pathway or produce knock outs of enzymes that are found to be central in this myo-inositol pathway, in order to demonstrate that the enzymes in question are essential for the myo-inositol conversion. Furthermore, I would suggest to set up a chromatographical method for the spent media and observe fermentations over time course. This would complement the NMR data, provide reasoning for the selected time points in the NMR experiments (which provide more in-depth analysis), and also be used for relevant control experiments.

Thank you for the suggestions that were taken into account – we set up a LC-MS/MS method for detecting intermediates and provided new data that support the proposed pathway.

The entire pathway proteins were among the most abundant proteins in the inositol condition and over-produced under inositol condition comparing to rhamnose condition that is a strong indication of the active involvement of these proteins in inositol fermentation. Nevertheless, we have tried to make knockouts of inositol pathway genes in *A. rhamnosivorans* but we failed to introduce foreign DNA in to the cells of this bacterium. Given the fact that this strain was relatively recently isolated

and characterized, it is highly challenging to engineer the genome using genome-editing tools. However, mother nature has helped us since *A. hadrus* DSM3319^T is a strain that does not have these genes and is not able to grow or metabolize *myo*-inositol (Fig. 1). So we can consider *A. hadrus* DSM3319 as a naturally occurring mutant lacking these pathway genes. Finally, as explained above all *myo*-inositol utilization genes are homologs of genes that have been shown to be essential for inositol degradation into 5-dehydro-2-deoxy-D-gluconate 6-phosphate in other bacteria, such as *Bacillus subtilis* (Yoshida *et al.*, 2008) and *Lactobacillus casei* (Yebra *et al.*, 2007). Therefore, the authors think it will not provide greater insight to repeat this in the current manuscript.

To further complement our NMR analysis, we also performed LC-MS/MS to identify intermediates of *myo*-inositol pathway. As a result, we could detect scyllo-inosose, 3,5/4-trihydroxycyclohexa-1,2-dione, 5-dehydro-2-deoxy-D-gluconate, and 5-dehydro-2-deoxy-D-gluconate 6-phosphate as intermediates, supporting our the proposed pathway in Fig. 2-3. Finally, we have performed repeated enzyme assays for the key enzyme (3-oxoacid CoA transferase) and convincingly show its activity toward propionate-CoA (Fig. 4E-F). We consider this as biochemical evidence of key pathway gene.

As suggested by the reviewer, we have also included the ¹²C measurement of inositol and end metabolites in Fig. 2 as well as Supplementary Fig. 3. These ¹²C and ¹³C-inositol fermentation data convincingly show that inositol was completely converted to propionate, acetate, formate and CO₂ as end metabolites using the indicated metabolic route.

Line 386-389. Authors begin sentence with “Remarkably” quick adaptation to changes in nutrient supply is nothing new, and has been shown by groups like Martens *et al.* on time adaptation of gut microbiota to various dietary fibers.

Thank you. We agree that quick adaptation is not new but this is a nice example of how butyrogenic bacteria, one of key functional groups in the gut, can switch between completely different central metabolisms from butyrate production to propionate production pathways. To avoid confusion, we have omitted “remarkably” in the text of the revised version. Please see line 474.

Fig. 1

Authors does not comment on the fact that acetate is produced to a similar extent as the propionate, why?

Thank you for pointing this out. Indeed acetate was produced at a similar amount as propionate. While most intestinal bacteria can produce acetate, only a few microbial species that have been reported to produce propionate including *Coproccoccus catus*; *Veillonella*; *Megasphaera* and *A. soehngeni*. Acetate production route is well known via mainly conversion of acetyl-CoA while the route for propionate production described in the current manuscript has not been described elsewhere. Nevertheless the role of acetate in the gut has been suggested to involve in body weight control and insulin sensitivity (Hernández *et al.*, 2019). Therefore we have added acetate in the title and abstract and included a small discussion in the introduction of the revised version of the manuscript. Please see lines 66-69.

Fig. 2. Very nice and informative Fig.. However, what kind of control experiments are provided? Furthermore, there is a lack of complementary analyses to support the

NMR evidence. E.g. evidence supported by chromatography or similar, on both C13 enriched and C12 samples would strengthen this study (and is essential) considerably. It would also allow methodology that are easier accessible than (high)-field-NMR, and make the experiments easier to validate.

Thank you. As stated in the manuscript, all growth experiment was performed in bicarbonate buffered medium containing inositol as the only energy and carbon source. It's clearly shown in Fig. 1 that ^{12}C -inositol was only converted to mainly propionate and acetate by 3 *Anaerostipes* strains. In absence of inositol, nothing was produced (Supplementary Fig. 1). All these were used as control for ^{13}C measurement. As suggested by the reviewer, we have now included the ^{12}C measurement of inositol and end metabolites in Fig. 2 as well as Supplementary Fig. 3. Our data on ^{12}C and ^{13}C -inositol fermentation data convincingly shows that the bacteria convert inositol to propionate, acetate, formate and CO_2 as end metabolites using the indicated metabolic route.

Fig. 4.

a) I would have liked a phylogenetic tree displaying *Anaerostipes* spp in a wider context, to provide readers a broader overview. What about propionate producer from other Phyla??

Thank you and we addressed this. Only a few propionate-producing bacteria have been reported and many of these do not use propionate CoA transferase for propionate production but propionate kinase (Louis & Flint, 2016). Genomic analysis showed only 2 strains (*Megasphaera elsdenii* and *Coproccoccus catus*) have oxoacid CoA transferase/propionate CoA transferase. Therefore we have included these two sequences in the tree. Please see revised Fig. 4A.

b) Each measurement is conducted in triplicates, I understand this as technical replicates. What about the biological replicates?

In fact the experiments shown in Fig. 4 were performed with one biological replicate. As to address this issue raised by the reviewer, we decided to repeat the enzyme assay with a new biological duplicate and 5 time points during the exponential phase. The results showed a clear increase of propionate CoA transferase activity along with propionate production and inositol consumption. These have been included as Fig. 4E-F in the revised version.

c) For B-D) 2 timepoints is (very) meager. I would suggest to run spent media from cultures, to observe the utilization of myo-inositol during time-course, this would provide a higher time resolution.

Thanks and we agree with the reviewer. Hence, we performed another experiment with 5 time points during exponential phase. The results showed a clear increase of propionate CoA transferase activity along with propionate production and inositol consumption.

Reviewer #2 (Remarks to the Author):

This manuscript reports a novel pathway for inositol conversion into propionate by several strains of *Anaerostipes* spp., highlighting the potential beneficial effects of inositol stereoisomers and phytate on human metabolic health. The novel pathway was characterized by ^{13}C -NMR and proteogenomics.

The paper is well written the results are relevant and technically well-conducted. The main drawback is the numerous assumptions or potential scenarios ("likely", "suggest

that might”, “might be a potential”, “might be related”) that are not supported with direct experimental data.

We thank the reviewer for the compliments on our work. To strengthen our claims, we have performed additional experiments including stool incubation in presence or absence of *A. rhamnosivorans* in [¹³C₆] phytate; metagenomic analysis on healthy/pre-diabetes/diabetes population; animal study in which insulin-resistant mice on a western diet (high in fat and sucrose) (Koh *et al.*, 2020) were fed with live *A. rhamnosivorans* with or without *myo*-inositol and used heat-killed *A. rhamnosivorans* for comparison. We also performed LC-MS/MS to detect the intermediates of *myo*-inositol pathway. Finally we have removed assumptions that were not supported by our experimental evidence from the revised manuscript. All changes in the revised manuscript are marked in yellow.

Of note, in the elucidation of the *myo*-inositol fermentation pathway many relevant intermediates were not experimentally detected. The authors only focused on precursor and final products, elucidating (indirectly) the numerous intermediate reactions and metabolites (depicted in Fig 3A) from the proteogenomic analysis. This could be due to poor sensitivity of NMR or other biochemical reasons. There are more sensitive techniques, mainly mass spectrometry and isotopologue analysis, that could nicely complement C-NMR and isotopomer characterization. In my view, it would be important to detect and characterize the isotopic enrichment of, at least, 5-dehydro-2-deoxy-D-gluconate 6-phosphate, 3-oxopropionate and 3-oxopropionyl-CoA.

The authors thank the reviewer for constructive suggestions. Following this suggestion, we performed LC-MS/MS on the cell extracts at 6 time points (30min; 1h; 2h; 3h; 4h and 24h) after inoculation and the data have been included in the revised version as Supplementary Fig. 4. We were able to detect scyllo-inosose, 3,5/4-trihydroxycyclohexa-1,2-dione, 5-dehydro-2-deoxy-D-gluconate and 5-dehydro-2-deoxy-D-gluconate 6-phosphate as intermediates but neither 3-oxopropionate nor 3-oxopropionyl-CoA were detected. The first 3 intermediates accumulated already after 30 min and remained relatively at high level for several hours before gradually being used afterwards. Interestingly 5-dehydro-2-deoxy-D-gluconate 6-phosphate was only detected at 1h and 2h time points with low signal, indicating that this compound was very quickly metabolized inside the cells. This rapid consumption may explain the inability to detect the following intermediates, including 3-oxopropionyl-CoA and 3-oxopropionate. Another reason might be due to their short half-times that limit the mass spectrometry analysis. This is in a good agreement with the proteomic data of which 3-oxoacid CoA transferase was substantially produced (under *myo*-inositol condition as compared to rhamnose condition (Fig. 3A-B), leading to no accumulation of 3-oxopropionate and 3-oxopropionyl-CoA in the cells. We believe that our NMR and LC-MS/MS data are now sufficiently strong to confirm our proposed pathway (Fig. 2-3).

The same holds true for the physiological relevance in human gut: It is stated that there is cross-feeding between *B. longum* subsp. *infantis* and *A. rhamnosivorans* on phytate via *myo*-inositol interspecies-transfer, leading to an increased propionate production, however this is not supported by direct quantitative stable isotope labeling data using ¹³C-phytate. Similarly, it is stated that “human often consume phytate-containing plants such as brans and cereals, and hence *myo*-inositol is likely released by microbial activities in the large intestine and serves as substrate for intestinal bacteria”, however no in vivo evidence is provided, either by administering ¹³C-inositol or ¹³C-phytate to mice

(which also contains *Anaerostipes* spp in the gut microbiota) and detecting labeled propionate in feces and serum.

Thank you and we agree that the coculture experiment of *B.longum* subsp *infantis* and *A. rhamnosivorans* on phytate has limitations. In our new set of experiments (Fig. 5) we show that [¹³C]propionate was only detected from [¹³C₆]phytate by gut bacteria when *A. rhamnosivorans* was supplemented (Fig. 5B-C) while *A. rhamnosivorans* alone was not able to metabolize [¹³C₆]phytate (Supplementary Fig. 10). These results indicate cross-feeding between gut bacteria and *A. rhamnosivorans* via inter-species inositol transfer, leading to propionate formation from phytate.

We are grateful for the suggestion to test some of our hypotheses *in vivo* in a mice experiment as we have done now.

However, setting up an *in vivo* experiment using ¹³C inositol or ¹³C phytate in mice is difficult due to the limit of resources. Hence, we performed instead enrichment studies in [¹³C₆]phytate in presence or absence of *A. rhamnosivorans* using human stool of 2 healthy donors (Fig. 5 and Supplementary Fig. 11). Our NMR data showed that intestinal bacteria of these two donors are capable of degrading phytate to mainly acetate and butyrate in absence of *A. rhamnosivorans*. The qPCR data on the cell number of *A. rhamnosivorans* also confirms the absence of *A. rhamnosivorans* in the microbiota of the fecal donors (Fig. 5D and Supplementary Fig. 9). Importantly, [¹³C]propionate was only detected in the enrichment that was supplemented with *A. rhamnosivorans* (Fig. 5B). Notably, similar results were also observed from the phytate enrichment from the second fecal donor (Supplementary Fig. 11).

Our metagenomic analysis in prediabetic/diabetic population showed that propionate production gene cluster is negatively correlated with metabolic biomarkers such as fasting insulin and serum triglyceride (Fig. 6A-B-C). These data suggest potential health promoting impacts of *Anaerostipes rhamnosivorans* that link to propionate production from inositol or phytate. To provide *in vivo* evidence of this, we performed an animal study of which mice were administered with live or heat-killed *A. rhamnosivorans* in presence or absence of *myo*-inositol. We observed that mice fed with live *A. rhamnosivorans* had lower fasting glucose level as compared to mice fed with heat-killed *A. rhamnosivorans* after 6 week treatment (3 times per week) and the reduction of fasting glucose level became significant once *A. rhamnosivorans* was administered with *myo*-inositol. In addition, we found a significant negative correlation (P=0.03) between the ratio of cecal propionate:butyrate and fasting serum insulin levels in mice fed with *A. rhamnosivorans* plus *myo*-inositol (Fig. 6E), indicating that active inositol fermentation to propionate in the cecum might be associated with lowering insulin level. To conclude, we observe that supplementation of *A. rhamnosivorans* with *myo*-inositol in mice reduced fasting glucose level and this reduction might be via active fermentation of inositol by *A. rhamnosivorans*, leading to the production of beneficial propionate.

Overall, I'm in favor of publication in Nature Communications after the preliminary nature of some claims are supported with additional experimental data.

Thank you !

Reviewer #3 (Remarks to the Author):

In this study the authors examine a few strains of *Anaerostipes* in the presence of

myo-inositol, and determine that some species from this genus can produce propionate (Fig. 1). They then go into Figure/discussion that help build the hypothesis that these strains may uniquely express pathways that enable conversion of myo-inositol to propionate. The authors discuss how this could be biologically important given the effects shown by others for propionate/SCFAs. They further suggest that breakdown of phytate by distinct commensal bacteria may provide an additional inositol source. The ideas presented in the manuscript are interesting, and are supported, to some degree, with data. However, the study falls short of validating the ideas proposed with experimentation that directly tests the suggested hypotheses. Loss/gain of function studies and colonization experiments are needed to support the authors' suggested mechanisms and *in vivo* significance.

Some specific concerns:

1. The strains on the left of Fig. 1 exhibit increased formate and propionate, in combination with decreased inositol. Thus it is clear why the author interpret that the bacteria metabolize inositol to propionate. However, these key initial studies lack necessary controls- including how propionate levels compare in this time course of bacterial growth in the same media when no inositol is present. It's possible that the metabolite changes are independent and this can not be known without this control. In addition, do propionate levels remain elevated once inositol is depleted? If so, for how long. The time course should be extended.

Thank you for these comments. The experiment was performed in a bicarbonate buffered medium containing inositol as the sole carbon and energy source; hence all compounds produced during the growth were likely from added substrate (*myo*-inositol). However, we understand the point from the reviewer. Hence, we have included the control condition in which the bacteria were inoculated in the medium without inositol and monitored the growth and metabolite production. We saw no metabolites that were produced in this control by any of the strains. The results of these experiments have been included in the revised manuscript as Supplementary Fig. 1.

We also repeated the growth curves of all strains in the same medium containing *myo*-inositol as substrate in triplicate with more sampling points. These have been included in Fig. 1 of the revised version. The data clearly show that inositol was depleted after 24h incubation and the cultures were still incubated for another 24h up to 48h. After 24h, the concentration of all end metabolites remained constant after *myo*-inositol was finished.

2. The authors discuss the microbiome and the likely relevance of the pathway *in vivo*. However, the biologic relevance *in vivo* is not examined anywhere in the paper. Do these strains survive in the intestine? If so, mice should be colonized with a propionate producer (*A. rham*), and well as a non-producer (*A. hadreus*) and metabolite levels compared to germ-free mice (with and without inositol).

We are grateful for the suggestion to test some of our hypothesis *in vivo* in mice

Hence, we performed 2 trials with insulin-resistant mice fed a western diet (rich in fat and sucrose) (Koh *et al.*, 2020). A pilot study was included as to investigate the usefulness of a mouse model by determining the fate of live cells of *A. rhamnosivorans* when administered via oral gavage. The results show that *A. rhamnosivorans* persisted well in fecal samples of mice fed with a western diet. We subsequently performed a second trial to explore potential benefit of *Anaerostipes* in

glucose metabolism using *A. rhamnosivorans* as a model bacterium and the data have been included in the revised version as Fig. 6. For the pilot study, *A. rhamnosivorans* was given to the mice by oral gavage every other day for a week with a dosage of 10×10^9 and the fecal samples were collected daily for quantification of *A. rhamnosivorans*. The results show that *A. rhamnosivorans* was able to persist in the mice and remained at high level after 3 days of treatment regardless of diets (Supplementary Fig. 13). The main study was performed for 7 weeks, in which mice were fed with either live *A. rhamnosivorans* or heat-killed *A. rhamnosivorans* in absence or presence of *myo*-inositol (0.1mg/g body weight). We observed that mice fed with live *A. rhamnosivorans* had lower fasting glucose level as compared to mice fed with heat-killed *A. rhamnosivorans* after 6 week treatment (3 times gavage per week) and the reduction of fasting glucose level became significant once *A. rhamnosivorans* was administered with *myo*-inositol. In addition, we found a significant negative correlation ($P=0.03$) between the ratio of cecal propionate:butyrate and fasting serum insulin levels for *A. rhamnosivorans* plus *myo*-inositol (Fig. 6E), indicating that active inositol fermentation to propionate in the cecum might be associated with lowering insulin level. To conclude, we observe that supplementation of *A. rhamnosivorans* with *myo*-inositol in mice reduced fasting glucose level and this reduction might be via active fermentation of inositol by *A. rhamnosivorans*, leading to the production of beneficial propionate.

3. Fig. 3A depicts the inositol conversion pathway and *A. rhamnosivorans* produces proteins that would be predicted to enable this pathway. However, it's unclear how values compared to other strains of bacteria that are not predicted to utilize this pathway. Fig. 3C mostly demonstrates similarities across the strains. However, the assumption is that the 2nd row (*Myo*-inositol 2 dehydrogenase) is the key difference between the inositol to propionate producer (*A. rham*), versus a non-producer (*A. hadrus*). This is a very big leap. Potential genes driving this difference were discussed in the text, but they should be manipulated in order to test the hypothesis that these proteins are the molecular mechanisms underlying the phenotype.

There might have been some misunderstanding in Fig. 3A. The difference between the propionate producers (*A. rhamnosivorans*; *A. hadrus* PEL85; *A. caccae*) and non-propionate producer (*A. hadrus* DSM3319^T) is the entire gene set for *myo*-inositol utilization and propionate production (Fig. 3C; completely absent in *A. hadrus* DSM3319^T but present in all other strains, though in different configurations). Although we did not indicate the values in Fig. 3C, we used colors ranging from yellow to blue representing low similarity (20%) to high similarity (100%) as shown in the color bar. In this way, the reader can immediately see which genomes have or do not have the *myo*-inositol pathway genes. Based on this, it is also easy to see that genome of *A. hadrus* DSM3319^T does not have *myo*-inositol pathway because it lacks almost the entire pathway genes and this was demonstrated in the growth experiment. We did not consider *myo*-inositol 2 dehydrogenase (the second row) as a key difference between propionate producers versus non-propionate producers. Instead we predicted this enzyme to involve in *chiro*-inositol (an isomer of *myo*-inositol) utilization (please see lines 252-253 and 297-301 in the revised manuscript). Strains capable of fermenting *myo*-inositol but lack of this gene cannot use *chiro*-inositol and this was shown experimentally (Supplementary Fig. 6).

4. Despite extensive discussion of phytate, the only Fig. with phytate is Supplemental Fig. 6 which is quite weak data. Further, it is unclear why *A. rhamnosivorans* alone has high acetate and butyrate- this is not consistent with Fig. 1. In addition, the study lacks the key control comparing to media without phytate. Thus, it lacks evidence that any differences are due to phytate metabolism. This direction of the study seems

tangential and unnecessary. Co-colonization in vivo would be necessary to support proposed conclusions.

The authors thank the reviewer to pointing this out. The control experiment of *A. rhamnosivorans* grown in YCFA without phytate has been now included in the Supplementary Fig. 8 and it clearly shows that butyrate and acetate production were from medium components such as yeast extract or peptone. To provide additional evidence of the involvement of *A. rhamnosivorans* in phytate degradation, leading to production of propionate, we have performed an enrichment study in [¹³C₆] phytate using stool of 2 healthy donors in presence or absence of *A. rhamnosivorans* (Fig. 5). Our NMR data showed that intestinal bacteria are capable of degrading phytate to mainly acetate and butyrate in absence of *A. rhamnosivorans*. Importantly, [¹³C]propionate was only detected in the enrichment where *A. rhamnosivorans* was supplemented (Fig. 5B). The qPCR data on the cell number of *A. rhamnosivorans* also confirmed the absence of *A. rhamnosivorans* in the microbiota of the fecal donor (Supplementary Fig. 9 and Fig. 5D). Notably, similar results were also observed from the phytate enrichment from the second fecal donor (Supplementary Fig. 11). In conclusion, our data show convincingly that propionate was only released from phytate degradation in presence of *A. rhamnosivorans*.

5. There are no statistics shown in any Fig., or described.

The detailed statistic for proteomic experiment, metagenomic analysis and mouse experiments was described in the material and methods as well and P value was included in the legend of Fig. 3 and 6. Other growth experiment and activity assay were performed in triplicate and the mean values were shown with standard deviation in all Figures. This has been indicated in the legend of all Figures.

Minor.

Some inaccuracies between the Fig. referenced and order in the text were noted. The authors thank the reviewer for pointing this out. We have checked carefully in the revised version, all changes have been highlighted.

References

- Corrado F, D'Anna R, Di Vieste G, Giordano D, Pintaudi B, Santamaria A & Di Benedetto A (2011) The effect of myoinositol supplementation on insulin resistance in patients with gestational diabetes. *Diabetic Medicine* **28**: 972-975.
- Croze ML, Géloën A & Soulage CO (2015) Abnormalities in myo-inositol metabolism associated with type 2 diabetes in mice fed a high-fat diet: benefits of a dietary myo-inositol supplementation. *British Journal of Nutrition* **113**: 1862-1875.
- D'Anna R, Scilipoti A, Giordano D, Caruso C, Cannata ML, Interdonato ML, Corrado F & Di Benedetto A (2013) myo-Inositol supplementation and onset of gestational diabetes mellitus in pregnant women with a family history of type 2 diabetes: a prospective, randomized, placebo-controlled study. *Diabetes Care* **36**: 854-857.
- Hernández MAG, Canfora EE, Jocken JWE & Blaak EE (2019) The Short-Chain Fatty Acid Acetate in Body Weight Control and Insulin Sensitivity. *Nutrients* **11**: 1943.

Koh A, Mannerås-Holm L, Yunn N-O, Nilsson PM, Ryu SH, Molinaro A, Perkins R, Smith JG & Bäckhed F (2020) Microbial Imidazole Propionate Affects Responses to Metformin through p38 γ -Dependent Inhibitory AMPK Phosphorylation. *Cell metabolism* **32**: 643-653.e644.

Louis P & Flint HJ (2016) Formation of propionate and butyrate by the human colonic microbiota. *Environmental Microbiology* n/a-n/a.

Nordio M & Proietti E (2012) The combined therapy with myo-inositol and D-chiro-inositol reduces the risk of metabolic disease in PCOS overweight patients compared to myo-inositol supplementation alone. *Eur Rev Med Pharmacol Sci* **16**: 575-581.

Ortmeyer HK, Larner J & Hansen BC (1995) Effects of D-Chiroinositol Added to a Meal on Plasma Glucose and Insulin in Hyperinsulinemic Rhesus Monkeys. *Obesity Research* **3**: 605S-608S.

Pagliai G, Russo E, Niccolai E, *et al.* (2020) Influence of a 3-month low-calorie Mediterranean diet compared to the vegetarian diet on human gut microbiota and SCFA: the CARDIVEG Study. *European Journal of Nutrition* **59**: 2011-2024.

Pintaudi B, Di Vieste G & Bonomo M (2016) The Effectiveness of Myo-Inositol and D-Chiro Inositol Treatment in Type 2 Diabetes. *Int J Endocrinol* **2016**: 9132052-9132052.

Yebra MJ, Zúñiga M, Beaufils S, Pérez-Martínez G, Deutscher J & Monedero V (2007) Identification of a gene cluster enabling *Lactobacillus casei* BL23 to utilize myo-inositol. *Applied and environmental microbiology* **73**: 3850-3858.

Yoshida K-i, Yamaguchi M, Morinaga T, Kinehara M, Ikeuchi M, Ashida H & Fujita Y (2008) myo-Inositol Catabolism in *Bacillus subtilis*. *Journal of Biological Chemistry* **283**: 10415-10424.

Reviewers' Comments:

Reviewer #1:

Remarks to the Author:

The authors have made a very comprehensive job considering comments, and for the major part improved the study considerably.

I have one major remark though. The authors do not present any evident proteomic data from complex consortia, merely a more indirect proteogenomic analysis in Fig3. How is the proteomic expression in a consortium, like e.g. the enrichment samples with phytate connected to Fig. 5, Suppl.Fig.9-11? Do you see the same enrichment of these proteins in such a situation, when you have a competitive environment? And are there other proteins that are highly enriched or negatively correlated as an effect of additional myo-inositol as a carbon source? Such an experiment would support the claim (This myo-inositol pathway). Do the authors see multifold changes in other unknown proteins, which may imply existence of; a) other similar cascades, b) other proteins being positively/negatively correlated with the A.rhamno myo-inositol cascade. Such a presentation would provide a better overview, and also a better view of how potential secondary effects may be affected as an effect of the myo-inositol pathway being switched on. I'm thinking in the line similarly, as we have seen in a recent study in our group <https://www.scientificamerican.com/custom-media/the-surprising-complexity-of-nurturing-a-healthy-gut/> . In this respect you would see not only that your proteins in the cascade are switched on, but also how expression of other proteins may be affected as side effects from the myo-inositol pathway being active.

a few minor comments.

L36. I thought the use of terms like 'novel' is not accepted by the journal.

L111. It may be feasible to include a sentence on microbiota-host interaction related to phytate metabolism here, which will expand the impact by potentially attracting a wider readership, by Referring to a recent Nature paper by Wu et al. "Microbiota-derived metabolite promotes HDAC3 activity in the gut".

there are a number of language errors which should be cleaned up (I only mention very few of these below), like e.g.:

L. 283-84 "Since these ethanol and lactate production also took place" delete these or replace with both.

L367 is one of many examples which lack a determiner "of annotated genes" change to "of the annotated genes"

L378 "a same amount of butyrate was", change to "a similar amount of butyrate was"

L494 "a several gene clusters" delete a

Reviewer #2:

Remarks to the Author:

The authors have done a good job during the review process and I'm fully satisfied with the new results.

Reviewer #3:

Remarks to the Author:

The authors have modified text and added new data to address questions raised in the original review. Although the revised manuscript and rebuttal address some questions/comments, there are still important unanswered questions regarding the in vivo significance.

1. Authors state that mice fed with live A. rhamnosivorans had lower fasting glucose levels as compared to mice fed with heat-killed A. rhamnosivorans after 6 week treatment, and the reduction of fasting glucose level became significant once A. rhamnosivorans was administered with myo-inositol- the p values go from 0.09 to 0.04 in the 2 different studies, which is not that striking. Also, the mean values are similar with and without myo-inositol. Unfortunately, these

data are on separate graphs so were not compared directly. Weight and GTT data do not demonstrate positive effects and fasting insulin is not shown. Instead, the ratio of cecal propionate:butyrate to fasting serum insulin levels is presented. Implications of this are completely correlative at this stage. Data from these in vivo studies are preliminary and not very convincing.

2. Authors elected to not include or discuss comparison of *A. rhamnosivorans* colonization to colonization with the non-producer (*A. hadreus*)- which their analyses show lacks most myo-inositol pathway genes. This seems to be the ideal comparison. Comparison to germ-free or mice depleted of microbiota +/- inositol would also be important controls. These studies are needed to strengthen the hypothesis that *A. rhamnosivorans*, and its myo-inositol pathway, are critical in vivo. However, the difference linked to live *A. rhamnosivorans* colonization seems to be limited to a small change described in fasting glucose so the physiologic relevance based on this approach will still remain somewhat unclear.

3. Authors state that the growth experiment and activity assays were performed in triplicate and the mean values were shown with standard deviation. Why were statistical analyses not shown for these? Standard deviation bars are not included in bar graphs like those in Fig. 4C. If these data are means of biological duplicate and technical triplicates, then that should be possible.

Reviewer #1 (Remarks to the Author):

The authors have made a very comprehensive job considering comments, and for the major part improved the study considerably.

I have one major remark though. The authors do not present any evident proteomic data from complex consortia, merely a more indirect proteogenomic analysis in Fig3. How is the proteomic expression in a consortium, like e.g. the enrichment samples with phytate connected to Fig. 5, Suppl.Fig.9-11? Do you see the same enrichment of these proteins in such a situation, when you have a competitive environment? And are there other proteins that are highly enriched or negatively correlated as an effect of additional myo-inositol as a carbon source? Such an experiment would support the claim (This myo-inositol pathway). Do the authors see multifold changes in other unknown proteins, which may imply existence of; a) other similar cascades, b) other proteins being positively/negatively correlated with the *A.rhamno* myo-inositol cascade. Such a presentation would provide a better overview, and also a better view of how potential secondary effects may be affected as an effect of the myo-inositol pathway being switched on. I'm thinking in the line similarly, as we have seen in a recent study in our group <https://www.scientificamerican.com/custom-media/the-surprising-complexity-of-nurturing-a-healthy-gut/> . In this respect you would see not only that your proteins in the cascade are switched on, but also how expression of other proteins may be affected as side effects from the myo-inositol pathway being active.

The authors thank the reviewer for the complements on the revised manuscript and suggestions for following studies on the phytate consortium involving the inositol pathway. While we have experience with metaproteomics in human systems we also understand its complexity and limitations (see Kolmeder et al. 2015). Hence, we consider deep metaproteomics analysis to be outside of the scope of this manuscript. Nevertheless, we made an attempt using the biomass obtained from ¹²C phytate enrichments with and without *A. rhamnosivorans* that was performed in parallel with exactly same volumes and same inoculums. This is because with the limited source for ¹³C phytate, we could only perform the enrichments in small volumes and all materials were used for NMR, HPLC, qPCR analysis (see Fig. 5). Unfortunately, due to the still limited biomass and high complexity of the microbial communities, we could detect only around 100 *Anaerostipes*-like proteins in the duplicate conditions where *A. rhamnosivorans* was added. While we detected several *A. rhamnosivorans* proteins involved in the new inositol pathway, the low level of signal did not allow us to perform a comprehensive proteome analysis. In absence of this, we conclude that further analysis of similar cascades would not be adding further insight. However, we think that our NMR data are sufficient to prove the involvement of *A. rhamnosivorans* in microbial phytate degradation, since we only observed the production of ¹³C-propionate when *A. rhamnosivorans* was added (see Fig. 5). We now also add this sentence in the revised manuscript 'It would be interesting to study the influence of environmental factors on the expression of the inositol pathway in *Anaerostipes* spp. in the gut.'. Please see lines 501-506.

a few minor comments.

L36. I thought the use of terms like 'novel' is not accepted by the journal.

We agree. We replace "novel" by "inositol"

L111. It may be feasible to include a sentence on microbiota-host interaction related to phytate metabolism here, which will expand the impact by potentially attracting a wider

readership, by Referring to a recent Nature paper by Wu et al. "Microbiota-derived metabolite promotes HDAC3 activity in the gut".

Thank you. We have added a sentence citing this reference in the revised version. Please see lines 111-112

there are a number of language errors which should be cleaned up (I only mention very few of these below), like e.g.:

L. 283-84 "Since these ethanol and lactate production also took place" delete these or replace with both.

Thank you and all changed as suggested. Please see line 279

L367 is one of many examples which lack a determiner "of annotated genes" change to "of the annotated genes"

Added. Please see line 361

L378 "a same amount of butyrate was", change to "a similar amount of butyrate was"

Changed. Please see line 372

L494 "a several gene clusters" delete a

Has been deleted. Please see line 382

Reviewer #2 (Remarks to the Author):

The authors have done a good job during the review process and I'm fully satisfied with the new results.

Thank you.

Reviewer #3 (Remarks to the Author):

The authors have modified text and added new data to address questions raised in the original review. Although the revised manuscript and rebuttal address some questions/comments, there are still important unanswered questions regarding the in vivo significance.

1. Authors state that mice fed with live *A. rhamnosivorans* had lower fasting glucose levels as compared to mice fed with heat-killed *A. rhamnosivorans* after 6 week treatment, and the reduction of fasting glucose level became significant once *A. rhamnosivorans* was administered with myo-inositol- the p values go from 0.09 to 0.04 in the 2 different studies, which is not that striking. Also, the mean values are similar with and without myo-inositol. Unfortunately, these data are on separate graphs so were not compared directly. Weight and GTT data do not demonstrate positive effects and fasting insulin is not shown. Instead, the ratio of cecal propionate:butyrate to fasting serum insulin levels is presented. Implications of this are completely correlative at this stage. Data from these in vivo studies are preliminary and not very convincing.

We agree that the effect of *A. rhamnosivorans* is modest and it requires further work to optimize all experimental conditions. Our results should be considered as an indication that *A. rhamnosivorans* may modulate metabolic parameters while the dosage of the bacteria and myo-inositol have to be further optimized. Insulin measurements from the OGTT have been included as supplementary figure 14E-F in the revised version. We have also included a discussion on the limitation of the in vivo results in the revised manuscript. Please see lines 498-504.

2. Authors elected to not include or discuss comparison of *A. rhamnosivorans* colonization to colonization with the non-producer (*A. hadreus*)- which their analyses show lacks most myo-inositol pathway genes. This seems to be the ideal comparison. Comparison to germ-free or mice depleted of microbiota +/- inositol would also be important controls. These studies are needed to strengthen the hypothesis that *A. rhamnosivorans*, and its myo-inositol pathway, are critical *in vivo*. However, the difference linked to live *A. rhamnosivorans* colonization seems to be limited to a small change described in fasting glucose so the physiologic relevance based on this approach will still remain somewhat unclear.

The authors agree with the reviewer that above animal studies could be done to strengthen the hypothesis of *A. rhamnosivorans* and the myo-inositol pathway *in vivo*. However, it is also important to consider that the effects of *A. rhamnosivorans* in presence of the entire microbiome may be different from this in germ-free mice, therefore, the proposed approach might not be best studying the efficacy of the bacteria. The impact of myo-inositol should be studied in a range of doses to find optimal dosage in future studies. Although, the impact of inositol on glucose metabolism has been studied in mice and human, we have not found any reports on the intestinal microbiome. Therefore, we believe that our results on the use of the combination of myo-inositol and *A. rhamnosivorans* to improve glucose metabolism is a welcome addition to the mechanistic insight on how inositol supplementation would affect insulin sensitivity and explain the observed heterogeneity in human interventions. As indicated above we have included a discussion on the limitation of the *in vivo* results in the revised manuscript. Please see lines 498-504

3. Authors state that the growth experiment and activity assays were performed in triplicate and the mean values were shown with standard deviation. Why were statistical analyses not shown for these? Standard deviation bars are not included in bar graphs like those in Fig. 4C. If these data are means of biological duplicate and technical triplicates, then that should be possible.

Indeed we have stated in the rebuttal of the first revision as well as in the material and methods of the first revised version of the manuscript that data in Figure 4C was based on one biological replicate (so a duplicate experiment) and technical triplicate (so analyzed three times). We understand the drawback of this and hence performed additional experiments with biological duplicates and technical triplicates as shown in Fig.4D. To avoid confusion, we have added standard deviation bars in Fig.4C of the revised version.

To facilitate the revision of all changes in the manuscript, we also list all modifications as below:

Line 34: 'novel' was replaced by 'inositol'

Lines 42-43: 'a diet induced insulin resistant mouse model' has been replaced by 'western-diet fed mice'

Lines 111-112: A sentence has been added 'a phytate-derived metabolite from the microbial conversion has shown to promote HDAC3 activity in the gut⁴⁰'

Line 134: 'insulin-resistant' was removed.

Line 233: The entire proteome has been included as supplementary data 2

Line 279: 'these ethanol and lactate' has been replaced by 'both ethanol and lactate'

Line 361: 'the' was added before 'annotated genes'

Line 372: 'same' was changed to 'similar'

Line 423: 'insulin resistant' was removed

Line 430: add Supplementary Fig 14F. Remove 'however'

Line 430 -437: The sentences changed to "We noted a trend (P=0.09) towards lower fasting glucose after 6 weeks of treatment with live *A. rhamnosivorans* compared to heat-killed bacteria (Fig. 6C) and a significant (P=0.04) lower fasting glucose when the mice were also treated with myo-inositol (Fig. 6D). In addition, after 7 weeks of treatment we found a significant negative correlation (P=0.03) between the ratio of cecal propionate:butyrate and fasting serum insulin levels for *A. rhamnosivorans* plus myo-inositol (Fig. 6E), suggesting that active inositol fermentation to propionate in the cecum might be associated with lowering insulin level"

Lines 437: this sentence has been removed 'In conclusion, we observed that supplementation of *A. rhamnosivorans* with myo-inositol in mice reduced fasting glucose level and this reduction might be via active fermentation of myo-inositol, leading to the production of beneficial propionate.'

Line 481: 'a several gene clusters' changed to 'several gene clusters'

Lines 501-506: a discussion on the animal work has been added as 'However, the unchanged glucose tolerance implies that further studies are needed to identifying the optimal treatment frequency, duration and dosage, as well as understanding the interactions between *A. rhamnosivorans* and host gut microbiota. Thus, it would be interesting to study the influence of environmental factors on the expression of the inositol pathway in the gut with a complete microbiome.'

This sentence has been removed 'Notably, the significant negative correlation between the ratio of cecal propionate:butyrate with fasting insulin in myo-inositol administered groups implies the potential association of inositol fermentation to propionate to lowering insulin level in mice.'

Line 507: The sentence was rephrased to 'A previous study reported that the presence of a microbial genomic structural variant of *A. hadrus* SSC/2 carrying myo-inositol pathway genes (Supplementary Fig. 5) was associated with reduced host metabolic disease risk.'

This sentence was removed 'Taken together, our results imply the potential involvement of myo-inositol conversion to propionate in the reduction of metabolic disease risks in human.'

Lines 511-512: The sentence was rephrased to 'Our detailed genomic data analysis showed that the genomes of *A. hadrus* type strain DSM 3319^T and *A. hadrus* DSM108065 do not predict the capacity to utilize myo-inositol, which was confirmed by our growth experiments.'

Line 516: 'experimentally' was added

Line 523: 'that' was added

Line 527: 'in this region' was changed to 'at this site'

Line 528: "as substrate for bacterial fermentation" was added.

Lines 828-831: This sentence was added 'Tail blood was also collected for insulin analysis (at t=-30m, 15 and 30 min) and analyzed with the Ultra-Sensitive Mouse Insulin ELISA kit (Crystal Chem, Downers Grove, IL, USA) according to the manufacturer's protocol.'

Line 1168: This sentence was added 'and shown values are means of technical replicates'.

References

Kolmeder, C.A., Ritari, J., Verdam, F.J., Muth, T., Keskitalo, S., Varjosalo, M., Fuentes, S., Greve, J.W., Buurman, W.A., Reichl, U., et al. (2015). Colonic metaproteomic signatures of active bacteria and the host in obesity. *Proteomics* 15, 3544-3552.

Michalak, L., Gaby, J.C., Lagos, L., La Rosa, S.L., Hvidsten, T.R., Tétard-Jones, C., Willats, W.G.T., Terrapon, N., Lombard, V., Henrissat, B., et al. (2020). Microbiota-directed fibre activates both targeted and secondary metabolic shifts in the distal gut. *Nature Communications* 11, 5773.

Reviewers' Comments:

Reviewer #1:

Remarks to the Author:

I thank the authors for addressing the issues raised by the reviewers. I still believe some solid proteomic data would provide a depth and add convincingly to the claim that this inositol pathway plays a central role also in more complex microbial systems.

Despite this, I think that in particular the NMR data in an elegant way demonstrates the pathway's metabolism in *A. stipedes* fermentations which warrants publication.

Best regards,
Bjørge Westereng.

Reviewer #3:

Remarks to the Author:

In my opinion, the new insulin data (supplemental 14E and 14F) show no difference in insulin levels and do not support the hypothesis that the authors want to suggest with the correlation data in Figure 6E. This may be why fasting insulin analyses were not originally included in the *A. rhamnosivorans* +/- myo-inositol studies, and why the lack of an insulin difference is not discussed in the current revised manuscript.

The final text of the results states that there is: "significant negative correlation ($P=0.03$) between the ratio of cecal propionate:butyrate and fasting serum insulin levels for *A. rhamnosivorans* plus myo-inositol (Fig. 6E), suggesting that active inositol fermentation to propionate in the cecum might be associated with lowering insulin level."

It's hard to justify this final statement when the evidence does not show decreased insulin levels with myo-inositol. I think the authors should at minimum indicate that no difference in insulin was observed with *A. rhamnosivorans* + myo-inositol (supplemental figure 14E, 14F) and discuss this in relation to the data shown in Figure 6E.

The other biochemical analyses are largely outside my expertise so I understand if you are accepting of the very preliminary nature of the *in vivo* work.

Reviewer #1 (Remarks to the Author):

I thank the authors for addressing the issues raised by the reviewers. I still believe some solid proteomic data would provide a depth and add convincingly to the claim that this inositol pathway plays a central role also in more complex microbial systems.

We thank the reviewer for the valuable suggestion and agree that further proteomic studies of complex microbial systems would be next steps as indicated in previous and final version of the manuscript. However, we like to point out that the observed association of metabolic biomarkers with the discovered inositol pathway in *Anaerostipes* spp. adds to the evidence for its potential beneficial contribution to metabolic health.

Despite this, I think that in particular the NMR data in an elegant way demonstrates the pathway's metabolism in Anaerostipes fermentations which warrants publication.

Thank you.

Best regards,
Bjørge Westereng.

Reviewer #3 (Remarks to the Author):

In my opinion, the new insulin data (supplemental 14E and 14F) show no difference in insulin levels and do not support the hypothesis that the authors want to suggest with the correlation data in Figure 6E. This may be why fasting insulin analyses were not originally included in the *A. rhamnosivorans* +/- myo-inositol studies, and why the lack of an insulin difference is not discussed in the current revised manuscript.

Thank you and we apologize for not inserting the original insulin data and only the ratios as below. We have now included the fasting insulin analyses as Supplementary Figure 14H and 14G (reproduced below). We see a decrease in fasting insulin in the treatments of live *A. rhamnosivorans* cells versus inactivated ones but no significance was reached. Despite the observed significant reduction of fasting glucose levels upon treatment of live bacteria and myo-inositol, the lack of significance in fasting insulin levels indicates the not-yet-optimal conditions of the animal study, including treatment duration, treatment frequency, and treatment dosage. The treatment duration may be an important factor since in a similar trial with diet-induced obese mice, another an intestinal butyrate-producing bacterium only improved insulin sensitivity after 13 and not 6 weeks (Le Roy et al., GUT 2021). This is now being discussed in the text of a revised version. Please see lines 432-439; 503-507 or below

In this mouse model we subsequently administered live or heat-killed cells of *A. rhamnosivorans* in the presence or absence of myo-inositol for 7 weeks. Body weight gain did not differ between treatment groups (Supplementary Fig. 14A-B) and neither did glucose tolerance determined after 6 weeks treatment (Supplementary Fig. 14C-F). We noted a trend ($P=0.09$) towards lower fasting glucose after 6 weeks of treatment with live *A. rhamnosivorans* compared to heat-killed bacteria (Fig. 6C) and a significant ($P=0.04$) lower fasting glucose when the mice were also treated with myo-inositol (Fig. 6D). In addition, the reduction of insulin levels was not significant between heat-killed and live treatment groups (Supplementary Fig. 14G-H) but we found a significant negative correlation ($P=0.03$) between the ratio of cecal propionate:butyrate and fasting serum insulin levels for treatments of only co-administration of myo-inositol with either live or heat-killed *A. rhamnosivorans* (Fig. 6E). This suggests that microbial inositol conversion to propionate rather than production of butyrate from carbohydrates or lactate and acetate may be

needed to decrease insulin sensitivity. As we did not observe a strong effect of active treatment further investigations are required.

Although no difference in neither insulin level nor glucose tolerance was observed, it has been shown that some intestinal bacteria need a much longer treatment with a high frequency in order to get the best effect while no effect was observed in a short treatment⁶³. Therefore, our results provide a first indication of potential health benefits of *A. rhamnosivorans* and implies that further studies are needed to identifying the optimal treatment frequency, duration and dosage, as well as understanding the interactions between *A. rhamnosivorans* and host gut microbiota.

Supplementary Figure 14H and G: Fasting insulin after 6 weeks treatment.

The final text of the results states that there is: "significant negative correlation ($P=0.03$) between the ratio of cecal propionate:butyrate and fasting serum insulin levels for *A. rhamnosivorans* plus myo-inositol (Fig. 6E), suggesting that active inositol fermentation to propionate in the cecum might be associated with lowering insulin level." Its hard to justify this final statement when the evidence does not show decreased insulin levels with myo-inositol. I think the authors should at minimum indicate that no difference in insulin was observed with *A. rhamnosivorans* + myo-inositol (supplemental figure 14E, 14F) and discuss this in relation to the data shown in Figure 6E.

We agree that this statement needs further nuances. Please note that treatment of the western-diet mice with the live *A. rhamnosivorans* resulted in a decreased fasting glucose levels that became significant upon addition of myo-inositol. We have now rephrased this in the revised version as 'In addition, the reduction of insulin levels was not significant between heat-killed and live treatment groups (Supplementary Fig. 14G-H) but we found a significant negative correlation ($P=0.03$) between the ratio of cecal propionate:butyrate and fasting serum insulin levels for treatments of only co-administration of myo-inositol with either live or heat-killed *A. rhamnosivorans* (Fig. 6E). This suggests that microbial inositol conversion to propionate rather than production of butyrate from carbohydrates or lactate and acetate may be needed to decrease insulin sensitivity. As we did not observe a strong effect of active treatment further investigations are required."

As indicated above, we have indicated in the text that no difference in insulin level was observed in the text and discussed the relation to Figure 6E. Please see lines 432-439; 503-507 (see above)

The other biochemical analyses are largely outside my expertise so I understand if you are accepting of the very preliminary nature of the *in vivo* work.

We thank the reviewer for being critical on our animal work and agree that observed *in vivo* effects are preliminary and need further investigations. However, we should not underestimate the results from the metagenomic analyses that provide strong correlations between propionate production genes and metabolic biomarkers (fasting insulin and triglyceride) in human populations. Our results are supported by previous findings that showed strong negative associations between a genomic structural variant carrying inositol fermentation genes (erroneously linked to butyrate production) and disease risk factors in Israeli cohort (Zeevi et al., *Nature* 2019). This has been included now on the revised manuscript. The *in vivo* experiments we describe here provide a preliminary indication for the validity of our hypothesis that the combination of inositol and active cells of an inositol-utilizing *Anaerostipes* species may improve insulin sensitivity and obviously requires further optimization to obtain the highest efficacy. This has been also indicated throughout the revised version of the manuscript.

References

- Zeevi, D.; Korem, T.; Godneva, A.; Bar, N.; Kurilshikov, A.; Lotan-Pompan, M.; Weinberger, A.; Fu, J.; Wijmenga, C.; Zhernakova, A.; Segal, E., Structural variation in the gut microbiome associates with host health. *Nature* **2019**.
- Le Roy, T.; Moens de Hase, E.; Van Hul, M.; Paquot, A.; Pelicaen, R.; Régnier, M.; Depommier, C.; Druart, C.; Everard, A.; Maiter, D.; Delzenne, N. M.; Bindels, L. B.; de Barse, M.; Loumaye, A.; Hermans, M. P.; Thissen, J.-P.; Vieira-Silva, S.; Falony, G.; Raes, J.; Muccioli, G. G.; Cani, P. D., *Dysosmobacter welbionis* is a newly isolated human commensal bacterium preventing diet-induced obesity and metabolic disorders in mice. *Gut* **2021**, gutjnl-2020-323778